∂ | **Open Peer Review** | *Clinical Microbiology* | Research Article

# *Candidatus* Methanosphaera massiliense sp. nov., a methanogenic archaeal species found in a human fecal sample and prevalent in pigs and red kangaroos

Virginie Pilliol,[1,2] Madjid Morsli,[1,3] Laureline Terlier,[1] Yasmine Hassani,[1,3] Ihab Malat,[1,3] Cheick Oumar Guindo,[1,3] Bernard Davoust,[1,3] Benjamin Lamglait,[4] Michel Drancourt,[1,3] Gérard Aboudharam,[1,2] Ghiles Grine,[3] Elodie Terrer[1,3]

**ABSTRACT** *Methanosphaera stadtmanae* was the sole *Methanosphaera* representative to be cultured and detected by molecular methods in the human gut microbiota, further associated with digestive and respiratory diseases, leaving unknown the actual diversity of human-associated *Methanosphaera* species. Here, a novel *Methanosphaera* species, *Candidatus* Methanosphaera massiliense (*Ca*. M. massiliense) sp. nov. was isolated by culture using a hydrogen- and carbon dioxide-free medium from one human feces sample. *Ca*. M. massiliense is a non-motile, 850 nm Gram-positive coccus autofluorescent at 420 nm. Whole-genome sequencing yielded a 29.7% GC content, gapless 1,785,773 bp genome sequence with an 84.5% coding ratio, encoding for alcohol and aldehyde dehydrogenases promoting the growth of *Ca*. M. massiliense without hydrogen. Screening additional mammal and human feces using a specific genome sequence-derived DNA-polymerase RT-PCR system yielded a prevalence of 22% in pigs, 12% in red kangaroos, and no detection in 149 other human samples. This study, extending the diversity of *Methanosphaera* in human microbiota, questions the zoonotic sources of *Ca*. M. massiliense and possible transfer between hosts.

**IMPORTANCE** Methanogens are constant inhabitants in the human gut microbiota in which *Methanosphaera stadtmanae* was the only cultivated *Methanosphaera* representative. We grew *Candidatus* Methanosphaera massiliense sp. nov. from one human feces sample in a novel culture medium under a nitrogen atmosphere. Systematic research for methanogens in human and animal fecal samples detected *Ca*. M. massiliense in pig and red kangaroo feces, raising the possibility of its zoonotic acquisition. Host specificity, source of acquisition, and adaptation of methanogens should be further investigated.

**KEYWORDS** methanogens, *Methanosphaera*, *Candidatus* Methanosphaera massiliense, hydrogen-free culture, human gut microbiota, host transfer

Human gut methanogens comprised of cultured representatives for *Methanobrevibacter smithii* (*M. smithii*) (1), *Methanobrevibacter oralis* (*M. oralis*) (2), *Methanobrevibacter arboriphilicus* (3), *Methanomassiliicoccus luminyensis* (*M. luminyensis*) (4), and *Methanosphaera stadtmanae* (*M. stadtmanae*) (5), *Candidatus* Methanomethylophilus alvus (6), *Candidatus* Methanomassiliicoccus intestinalis (7), and *Candidatus* Methanobrevibacter intestini (8–10), with a combined prevalence of 97.5% for *M. smithii* and 30% for *M. stadtmanae* (11) which indicated these various methanogens were overlapping constant gut inhabitants (12). An exception is the dysbiotic depletion of *M. smithii* and its absence in children with deadly severe acute malnutrition (13).

Since the detection of *Methanosphaera* spp. and *Methanobrevibacter* spp. in the bioaerosols from poultries, dairy farms, and piggeries (14–16), studies were conducted

Address correspondence to Elodie Terrer, elodie.terrer@univ-amu.fr, or Ghiles Grine, grineghiles@gmail.com.

V.P., M.D., G.A., G.G., and E.T. are co-inventors of a patented culture medium referred to as GG culture medium in this report (patent FR 23 01404).

See the funding table on p. 13.

in animal models to investigate the role of methanogens in hypersensitivity diseases. The administration of *M. stadtmanae* extract aerosol but not *M. smithii* aerosol in the airways of mice yielded a hypersensitivity response with an accumulation of eosinophils and neutrophils in the lungs (17, 18), laying the groundwork for the potential pathogenic role of *M. stadtmanae*. The pro-inflammatory capacity of *M. stadtmanae* was next used to induce hypersensitivity pneumonitis disease in mice models (19, 20). Moreover, further studies focused on human cells' response exposition to methanogens, *M. smithii* and *M. stadtmanae* are both able to activate mononuclear cells but *M. stadtmanae* induced a higher release of pro-inflammatory cytokines than *M. smithii* (21, 22). *M. stadtmanae* phagocytosis is crucial for cell activation (21). At the molecular level, *M. stadtmanae* RNA was identified as the microbe-associated molecular pattern (MAMP) that can trigger an NLRP3 inflammasome activation through the toll-like receptor 8 (TLR8) (23). Interestingly, *M. stadtmanae* was more prevalent in patients with inflammatory bowel syndrome which developed a specific IgG response to this methanogen (22). More recently, a decrease in the *Methanosphaera* abundance was noticed in healthy to long-time diabetic patients, whereas an increase in *M. smithii* was observed (24).

Although *M. stadtmanae* was for a long time the sole *Methanosphaera* member cultured from human feces (5) and further detected by PCR-based methods in human dental plaque (25), a new human isolate, *Methanosphaera* sp. PA5 was provided by Hoedt et al. in 2018 (26). Furthermore, *Methanosphaera* has been cultured from mammal guts including *Methanosphaera cuniculi* (*M. cuniculi*) in rabbits (27), *Methanosphaera* sp. WGK6 in Western grey kangaroo (*Macropus fuliginosus*) (28), and *Methanosphaera* sp. BMS in cow guts (26). Recently, Hoedt et al. (26) and Chibani et al. (9) obtained, respectively, 7 and 20 *Methanosphaera* spp. genomes from metagenomic data of mammals (including humans), some of them not corresponding to known isolates. Particularly, among the 20 *Methanosphaera* spp. genomes provided by Chibani et al. from the human gut, 17 were related to *M. stadtmanae*, two to *M. cuniculi*, and one to a genome obtained from cow rumen, suggesting possible host transfers and adaptation from animals to human gut.

These cumulative data suggested that human health interest, host-associated, and yet-uncultured *Methanosphaera* members remained to be discovered by isolation and culture in the human gut microbiota, subject to the opportunity to apply renewed culture approaches to suitable clinical materials. Using a recently designed hydrogen-free and carbon dioxide-free medium named GG medium (29, patent FR 23 01404), combined with molecular approaches, our investigations of mammal and human feces led to the discovery of a novel *Methanosphaera* representative isolate.

## RESULTS

### Isolation and microscopic observations

Methane was detected from one methanogen 16S rRNA gene PCR-positive stool sample culture. Successive transfers and subcultures on a solid medium yielded colonies forming a translucent carpet (Fig. S1), which was further subcultured in a liquid medium and produced methane after a 15-day incubation at 37°C. Colonies comprised of Gram-positive diplococci and clumps, with the individual diameter measured at 850 nm using electron microscopy. Gram variability was observed after 15 days of culture. These non-motile cocci showed auto-fluorescence at 420 nm using confocal microscopy. The purity of the final culture was established as only autofluorescent cocci were observed in confocal microscopy from several fields as well and only cocci were observed in electron microscopy. Furthermore, we checked that no bacteria grew on a COS agar plate (Fig. 1)

### Genome assembly and analysis

The methanogen 16S rRNA gene sequence determined after PCR sequencing exhibited 100% coverage and 95.77% identity with the homologous sequence of *M. cuniculi* (NCBI accession number: NR_104874.1), and 100% coverage and 95.6% identity with the homologous sequence of *M. stadtmanae* (NCBI accession number: NR_074323.1),

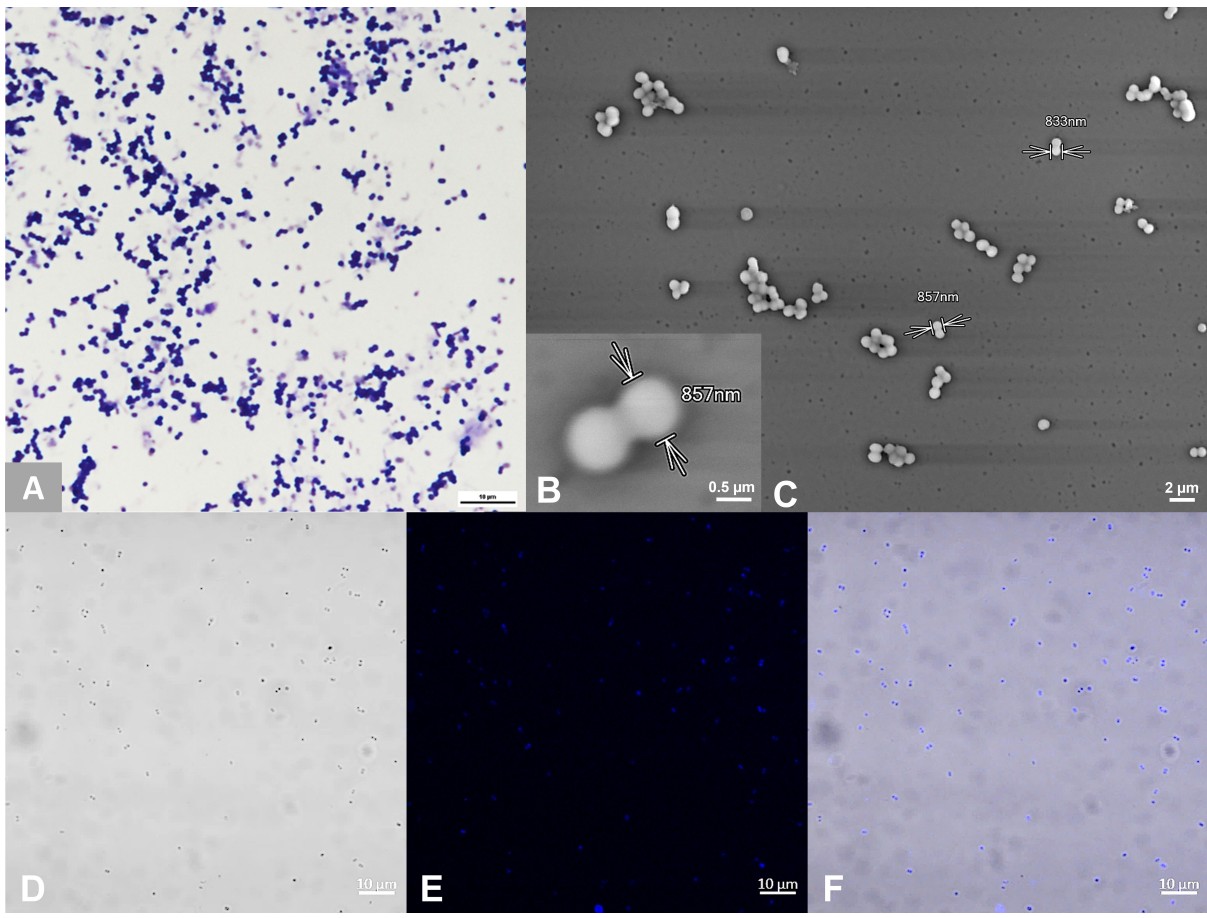

**FIG 1** Microscopy features of *Candidatus* methanosphaera massiliense sp. nov. (A) Gram coloration, *Ca*. M. massiliense appears as Gram-positive cocci, diplococci, or clump. (B) Electron microscopy (TM4000 HITACHI, 10 KV, ×10,000), zoom on *Ca*. M. massiliense diplococcus. (C) Electron microscopy (TM4000 HITACHI, 10 KV, ×2,500), showing a large view of *Ca*. M. massiliense culture, the methanogen appears as cocci, diplococci, and clumps. (D) Confocal microscopy, brightfill mode view of *Ca*. M. massiliense culture (LSM 900 [Carl Zeiss Microscopy GmbH]). (E) Confocal microscopy, autofluorescence at 420 nm: all the cocci, diplococci, and clumps are autofluorescent (LSM 900 [Carl Zeiss Microscopy GmbH]). (F) Confocal microscopy merged view of autofluorescence at 420 nm and bright fill mode (LSM 900 [Carl Zeiss Microscopy GmbH]).

suggesting a novel *Methanosphaera* species whose genome was then sequenced. The combined assembly of Illumina and Nanopore data generated two contigs totaling 1,785,773 base pairs (bp) close to the 1,767,403 bp genome sequence size of *M. stadtmanae* (NCBI accession number: GCA_000012545.1) whose dDNA-DNA Hybridization (dDDH) was 23.5%. CheckM completeness was 97.6% without contamination. Genome annotation yielded an 84.5% coding ratio for 1,690 protein-encoding genes with 1,232 assigned to Clusters of Orthologous Groups (COGs) (Table S1), 42 tRNA-encoding genes, 4 rRNA, and 2 CRISPRs (Table 1; Fig. 2). The nucleotide composition yielded 29.7% GC content with 0% gaps. The 16S rRNA gene sequence from the whole genome shared 97.22% sequence similarity (100% coverage) with *M. stadtmanae*, 96.95% (100% coverage) with *Methanosphaera* sp. BMS, and 96.55% (99% coverage) with *M. cuniculi* and 98.9% (92% coverage) with *Methanosphaera* sp. WGK6. The 16S rRNA sequence-based phylogenetic tree showed that the new methanogen clustered with only host-associated sequences and not with environmental ones, it also first clustered with uncultured archaeon clones from pig feces (accession n°: HM573447.1 and HM573413.1) and *Methanosphaera* sp. WGK6 16S rRNA (accession n°: KF697728.1) (Table S2; Fig. 3). Whole-genome sequence (WGS)-based phylogeny incorporating 37 genomes including *Methanosphaera* genomes recovered from Chibani et al. and Hoedt et al. (Table S3) indicated the methanogen was distant from the cultured species, *Methanopshaera* sp.

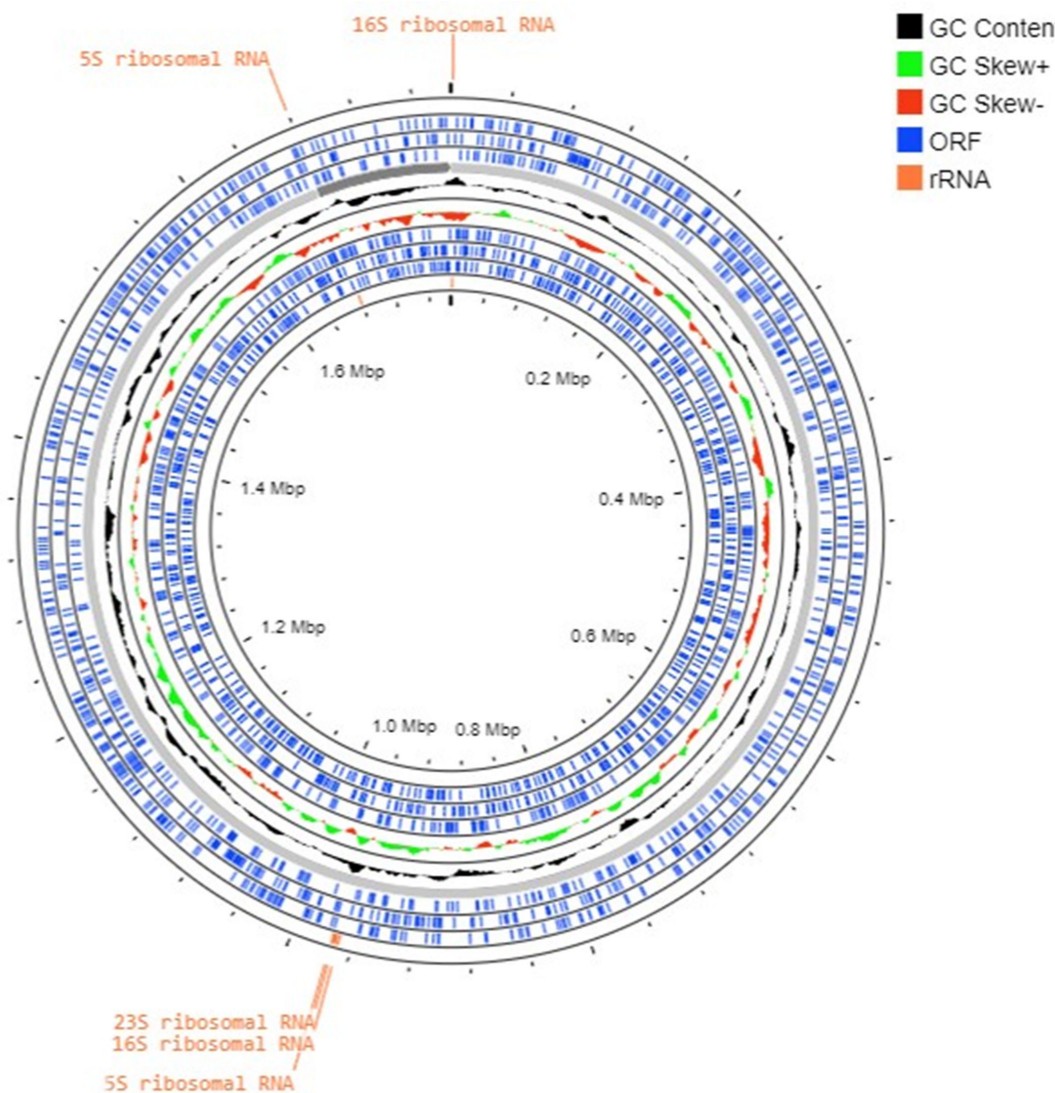

**FIG 2** Circular representation of *Candidatus* methanosphaera massiliense sp. nov. genome. Genome representation was performed using the Proksee platform with standard parameters (version 1.0.0) (https://proksee.ca/). The two contigs are presented with color-coded highlights to distinguish important genomic features and provide information about the composition and structure of the *Candidatus* Methanosphaera massiliense sp. nov. genome. ORF regions are indicated in blue, rRNA genes in orange, GC content in black, and positive and negative GC skew in green and red, respectively. This analysis reveals the distribution of protein-coding genes and the variation in GC content across the genome.

WGK6 (average nucleotide identity [ANI], 79%) isolated from Western grey kangaroo gut, followed by *M. stadtmanae* (ANI, 78%) isolated from human gut, *M. cuniculi* (ANI, 75%) isolated from rabbit gut and *Methanosphaera* sp. BMS, a bovine species (ANI, 75%) (Fig. 4). However, it clustered with *Methanosphaera* sp. RUG761 from bovine (ANI, 99%) and *Methanosphaera* sp. M5 from humans with colorectal cancer (ANI, 99%), two genomes that clustered together (ANI, 99%), and with *Methanosphaera* sp. SHI1033 from sheep (ANI, 97%) (Fig. 4). The dDDH values with these three species were above 70% whatever the DDH calculation method (Table S3). Following these phylogeny data and culture isolation, we introduced a new methanogen species named *Candidatus* Methanosphaera massiliense (*Ca.* M. massiliense) sp. nov. (strain: BB6), "massiliense" is for "Marseille," the city where it was isolated. It was deposited into the international public collection CSUR WDCM 875 (Collection Souches Unité des Rickettsies, WDCM 875, Marseille, France) under the reference Q7470 and sent to the Leibniz Institute DSMZ, German Collection of Microorganisms and Cell Cultures.

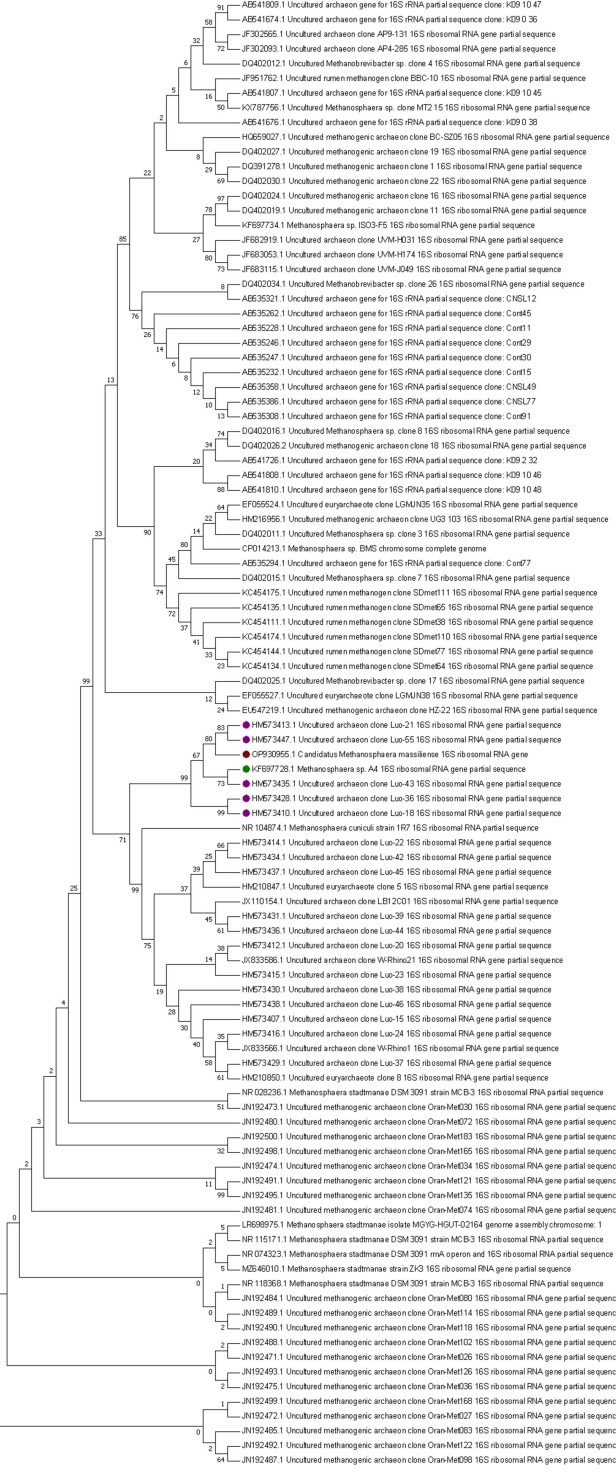

**FIG 3** Maximum likelihood phylogenetic tree based on 16S rRNA sequence analysis of *Candidatus* Methanosphaera massiliense sp. nov. and the first 100 hit blasts downloaded from NCBI GenBank database (03-01-2023). The 16S rRNA sequence-based phylogenetic tree showed that *Candidatus* Methanosphaera massiliense sp. nov. (red) first clustered with uncultured archaeon clones from pig feces (green) and *Methanosphaera* sp. from western grey kangaroo (purple).

## MALDI-TOF mass spectrometry

MALDI-TOF mass spectrometry yielded reproducible peptide spectra but no identification by spectra comparison with our bacterial database. The cell mass was first confirmed through PCR sequencing to match with BB6 isolate before MALDI-TOF analysis. Comparison with *M. stadtmanae* DSM3091 peptidic profile did not show significant differences within the two spectra (Fig. S2).

## Antibiotic susceptibility

Antibiotic susceptibility testing by macrodilution indicated that the methanogen was *in vitro* resistant to amoxicillin (minimum inhibitory concentration [MIC] > 100 mg/L), imipenem (MIC > 100 mg/L), fusidic acid (MIC > 4 mg/L), and bacitracin (MIC > 4 mg/L), and *in vitro* susceptible to metronidazole (MIC <1 mg/L) and chloramphenicol (MIC < 4 mg/L); while all antibiotics were active as expected against the appropriate *Escherichia coli* and *Staphylococcus aureus* controls (Table S4).

## pH susceptibilily

pH susceptibility testing showed that isolate BB6 is also able to grow at pH 6 and pH 7.3 but not out of this range. Subcultures from these two pH conditions yielded to methane production after a 7-day culture. The pH of 7.3 in the GG medium was considered the reference pH for the methanogen culture (Table S4).

## Substrate affinity testing and methanogenesis pathways

*Ca. M. massiliense* was able to grow on GG0 medium (basal GG medium) with methanol under $H_2/CO_2/N_2$ atmosphere, GG0 medium with methanol and ethanol under nitrogen atmosphere, and GG medium (containing acetate, formate, and methanol) supplemented with ethanol as the reference condition (Table S4). Proxi-growth curves of methane production and optical density from day 0 to day 7 are also available in Table S4. Slow growth was observed on methanol alone. No methane was detected with no substrate, under the $H_2/CO_2/N_2$ atmosphere and no other substrate, on formate, acetate, ethanol alone, or ethanol under the $H_2/CO_2/N_2$ atmosphere (Table S4). Genome annotation revealed the presence of a NADPH-butanol dehydrogenase and an aldehyde dehydrogenase closely related to *Methanosphaera* sp. SHI1033 and *Methanosphaera* sp. WGK6 (Figures S3a through d), which allows growth in a hydrogen-free atmosphere in a medium containing methanol and ethanol. The absence of formate dehydrogenase and the presence of the incomplete enzymatic pathway for acetoclastic methanogenesis, only the acetyl-CoA synthetase is encoded, are consistent with the previous experimental results.

**TABLE 1** *Candidatus* Methanosphaera massiliense sp. nov. genome sequence characteristics (accession number: JARBXM000000000)

| | |
|---|---|
| Total sequence length (bp) | 1,785,773 |
| Number of sequences | 2 |
| Longest sequences (bp) | 1,679,591 |
| N50 (bp) | 1,679,591 |
| Gap ratio (%) | 0 |
| GC content (%) | 29.7 |
| Number of CDSs | 1,690 |
| Average protein length | 297.6 |
| Coding ratio (%) | 84.5 |
| Number of rRNAs | 4 |
| Number of tRNAs | 42 |
| Number of CRISPRs | 2 |
| CheckM completeness (%) | 97.6 |
| CheckM contamination (%) | 0 |

## Feces screening by specific RT-PCR and amplicon sequencing

The specificity of the *Ca*. M. massiliense RT-PCR system was confirmed as all methanogen strains including *Methanobrevibacter* and *Methanosphaera* species and negative controls remained negative but was positive for *Ca*. M. massiliense, while the *Methanobrevibacter* and *Methanosphaera* strains yielded positivity for the 16S rRNA gene (Table S5). Moreover, the RT-PCR system was *in silico* positive for *Ca*. M. massiliense, *Methanosphaera* sp. Mb5 and RUG761 but not for *Methanosphaera* sp. SHI1033 and WGK6 nor *M. stadtmanae*. Then, a total of 150 leftover human stool samples and 313 ground-collected feces from six mammal species located in four French departments (Fig. S4) were investigated using a *Ca*. M. massiliense -specific RT-PCR system, with a Ct value <37 considered as positive. One of 150 human fecal samples (0.67%), 14/64 (22%) pig fecal samples, and 6/50 (12%) red kangaroo fecal samples were positive by RT-PCR for *Ca*. M. massiliense while the other mammal samples remained negative (Table S6a). The one

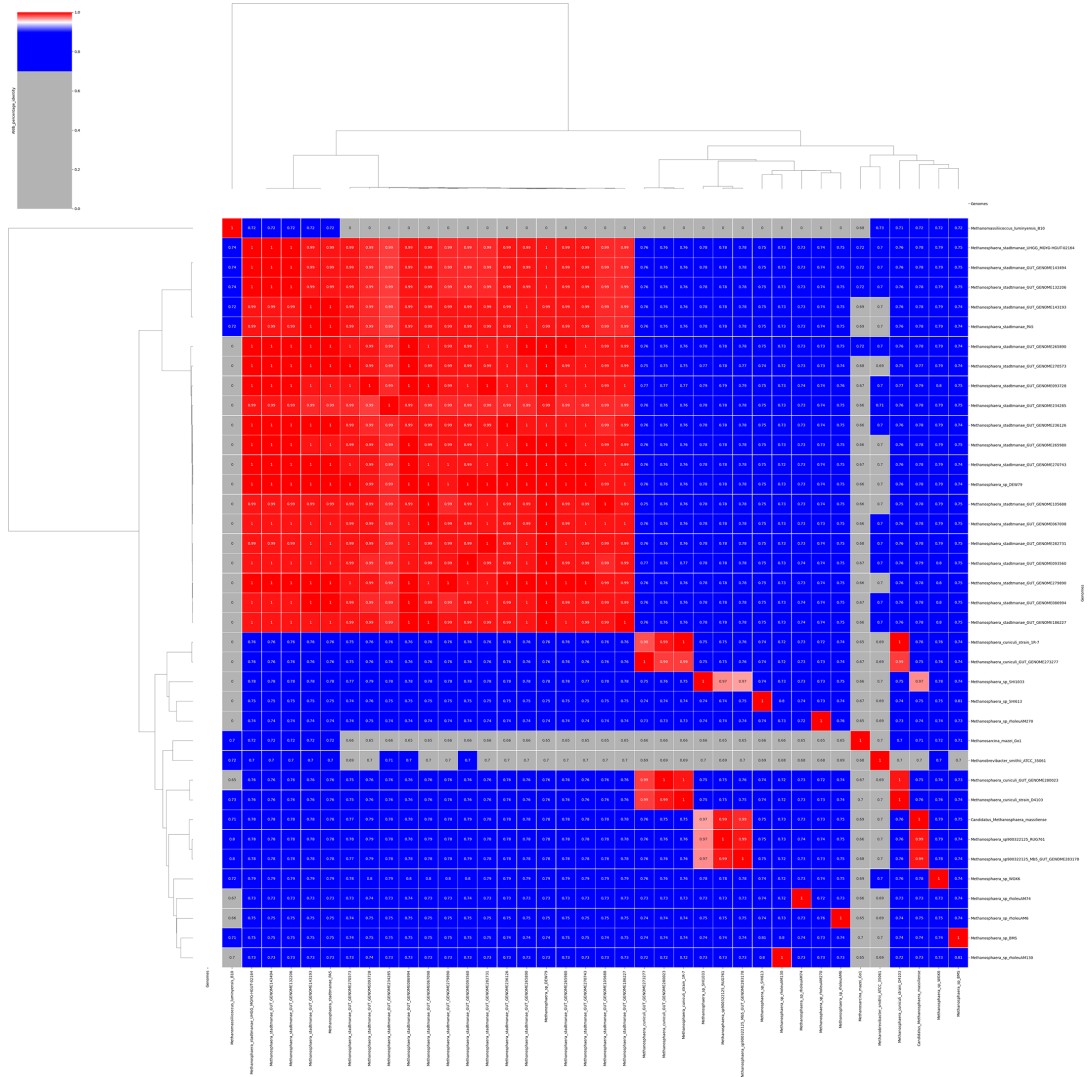

**FIG 4** Heatmap phylogeny of *Candidatus* Methanosphaera massiliense sp. nov. and closely related methanogen species generated with ANI values calculated using PYANI software version (0.2.7) with standard parameters. *Ca*. M. massiliense was distant from the cultured *Methanosphaera* species, *Methanopshaera* sp. WGK6 (Average Nucleotide Identity (ANI), 79%) isolated from Western grey kangaroo gut, followed by *M. stadtmanae* (ANI, 78%) isolated from human gut, *M. cuniculi* (ANI, 75%) isolated from rabbit gut and *Methanosphaera* sp. BMS, a bovine species (ANI, 75%) but it clustered with *Methanosphaera* sp. RUG761 (ANI, 99%) detected in bovine and *Methanosphaera* sp. M5 (ANI, 99%) obtained from a man with colorectal cancer, the two genomes clustered together (ANI, 99%), and with *Methanosphaera* sp. SHI1033 (ANI, 97%), from sheep.

positive human feces sample was the same as *Ca*. M. massiliense had been isolated by culture as above. *Ca*. M. massiliense identity in all positive samples was confirmed as the sequences showed an identity average of 96.28% ± 0.82 and 100% coverage with *Ca*. M. massiliense whole-genome sequence and no other correspondence was found in the GenBank database (Table S6b). Moreover, *Ca*. M. massiliense load was comparable in the human sample (3,77E+07 CFU/mL) and in pig and kangaroo samples (average load 2.68E+07 CFU/mL ± 1.25E+07) (Table S6a).

## DISCUSSION

*Ca*. Methanosphaera massiliense is the second *Methanosphaera* species isolated in humans and described using a polyphasic approach combining morphology by light and electron microscopy, peptide fingerprinting by mass spectrometry, antibiotics, pH susceptibility, substrate affinity testing, and whole-genome sequence-derived analyses, with the negative results of controls introduced in all experiments confirming the authenticity of the results.

Whole-genome sequence and 16S rRNA gene sequence data confirmed the originality of *Ca*. M. massiliense's identity among isolated species but *Ca*. M. massiliense might be the cultured representative of *Methanosphaera* sp. RUG761 and *Methanosphaera* sp. M5, a methanogen detected in bovine and a man with colorectal cancer (9).

Among isolated species, *Ca*. M. massiliense was found to be most closely metabolically related to *Methanosphaera* sp. WGK6, a methanogen isolated by culture in one Western grey kangaroo (28). Both methanogens can grow on methanol and hydrogen but also in a hydrogen-free medium containing methanol and ethanol, as they share NADPH-butanol dehydrogenase and aldehyde dehydrogenase (28), GG medium supplemented with ethanol should be suitable for growing these methanogens. A distinctive feature of *M. stadtmanae*, *M. cuniculi,* and *Methanosphaera* sp. BMS is that they have only a hydrogen-dependent methanol-fueled metabolism (26, 27, 30).

The enzymes identified through genome sequencing and annotation play a crucial role in characterizing the population susceptible to the presence of *Ca*. M. massiliense in the gut microbiota. This population may consist of individuals who have had exposure to ethanol and methanol, both of which can originate from either bacterial and fungal sources or consumable items like beverages and food products (31–33). Indeed, ethanol and methanol are both chemical compounds found in various sources that can be consumed by humans. Ethanol is primarily found in alcoholic beverages such as beer, wine, and spirits (31). In addition, ethanol can be used as an additive or solvent in certain foodstuffs (32). Methanol can be present in foods due to natural fermentation processes or the breakdown of pectin in certain fruits and vegetables (33). Alcoholic beverages containing ethanol may also contain low levels of methanol, although these are generally considered safe for human consumption (34, 35). Also, the fact that *Ca*. M. massiliense only remained alive in a narrow neutral pH further suggesting that individuals lacking gastric acidity may favor the implantation of *Ca*. M. massiliense in their gut, as well as individuals exposed to large *Ca*. M. massiliense inoculum (36).

Considering the findings from this study, we propose a hypothesis suggesting that potential sources of *Ca*. M. massiliense may be of mammalian zoonotic origin, as indicated by the observed prevalence of *Ca*. M. massiliense in pigs destined to slaughterhouse. This species has also been identified in other animals, including those with rumen, cow, sheep, and human hosts (9). Fundamentally, our research suggests that this methanogenic archaeal species is not confined to a single host type and does not exhibit host-specific lineages. This implies that food consumption and other contact with animals could serve as pathways for the transmission of these microorganisms to the human intestinal microbiota (37). The success of crossing implantation in individuals of other mammal species, including humans, may depend on the density of contact between species and individual-specific factors. Notably, other methanogens, particularly *Methanobrevibacter* spp. and *Methanomassiliicoccus* spp., have been detected in both humans and other mammals (38–40), such as pigs: *M. smithii*, *Methanobrevibacter*

*millerae* (*M. millerae*), and *M. luminyensis* (38). This finding underscores the intricate nature of microbial interactions within diverse host environments, emphasizing the need for additional research to comprehend the dynamics of microbiota exchange and its potential implications for human and animal health.

In conclusion, we are reporting the second isolation by culture of a *Methanosphaera* species from human feces referred to as *Ca*. M. massiliense sp. nov., which was obtained using a home-made specific culture GG medium under a nitrogen atmosphere. It is genetically more closely related to mammalian animals *Methanosphaera* than to *M. stadtmanae*. Our findings raise the question of potential zoonotic sources and the conditions of adaptation to a minority of individuals of *Ca*. M. massiliense and methanogens in general and their possible consequences to human health.

## MATERIALS AND METHODS

### Specimen collection

This prospective study was conducted from August 2021 to September 2021 at the Institut Hospitalo-Universitaire (IHU) Méditerranée Infection in Marseille, France. In this study, only leftover stool samples were studied after the microbiological diagnosis laboratory work was completed and after the investigators had verified that no patient objected to the use of leftover clinical samples for research purposes (Article L1211-2 of the French Public Health Code). According to French law, the research on anonymized leftover stool samples used in this study was considered non-interventional research (Article L1221-1.1 of the French Public Health Code) and did not require any ethics committee approval. In addition, we used animal fecal samples collected on the ground. No ethical approval was required for these samples, for which the mammal owners' oral consent was obtained by two of us (B.D. and B.L.). Samples were screened for methanogens presence using RT-PCR (primers: Metha_16S_2_MBF, 5′-CGAACCGGATTAGATACCCG-3′ and Metha_16S_2_MBR, 5′-CCCGCCAATTCCTTTAAGTT-3′; probe: FAM_Metha_16S_2_MBP, [FAM]-5′-CCTGGGAAGTACGGTCGCAAG-3′) as previously described (41).

### Culture and isolation of the new methanogen species

A 0.5 g 16S rRNA RT-PCR-positive stool sample was suspended in 10 mL of Dulbecco's phosphate-buffered saline (DPBS) 1× water (Thermo Fisher Scientific, Waltham, MA, USA) and vortexed for a few seconds. A 200 µL volume of this suspension was inoculated into a Hungate tube containing GG medium (29, patent FR 23 01404) that is derived from SAB medium (42), containing acetate, formate, and methanol as standards, previously degassed for 3 minutes with 2 bar nitrogen. The inoculated tube was incubated for 7 days at 37°C. On day 7, methane was detected by gas chromatography using a Clarus 580 chromatograph (Perkin Elmer, Waltham, MA, USA) confirming methanogen growth as previously described (42) and optical density was measured by putting the Hungate tubes in a spectrophotometer cell density meter model 40 (Thermo Fisher Scientific, Waltham, MA, USA). Then, 200 µL of growing culture was inoculated into a fresh GG medium containing fresh antibiotics, 100 mg/L amoxicillin, 100 mg/L vancomycin, 100 mg/L imipenem, 50 mg/L daptomycin, and 50 mg/L amphotericin B (Sigma Aldrich, Saint Quentin Fallavier, France) and supplemented with ethanol at 2% (vol/vol). Then, 100 µL of the growing culture was inoculated on the same home-made solid GG medium obtained by adding 15 g/L agar (Condalab, Madrid, Spain) to the broth formulation of the GG medium with ethanol which was then incubated at 37°C in an anaerobic atmosphere using BD Pouch Gaspak Ez Anaerobe System (Becton, Dickinson and Company, Franklin Lakes, USA) for 9 days. Negative controls consisted in the inoculation of a solid medium with PBS, the liquid medium previously inoculated with PBS for 7 days (Fig. 1). The isolate was sub-cultured 10 times to ensure the purity of the culture, checked by autofluorescence using a confocal light microscope LSM 900 (Carl Zeiss Microscopy GmbH, Jena,

Germany) at the 63× objective with immersion oil as previously described (43), electron microscopy with the Scanning Electron Microscope TM4000 plus (Hitachi, Tokyo, Japan), and inoculation of 100 µL on Columbia agar 5% sheep blood (COS) media (bioMérieux, Marcy-Étoile, France) plate.

## Light microscopy

Gram staining was performed as previously described (44). For confocal microscopy, a 10 µL volume from the pure culture was placed on a glass slide and sealed with a coverslip under an anaerobic chamber. Auto-fluorescence was observed under a confocal light microscope LSM 900 (Carl Zeiss Microscopy GmbH, Jena, Germany) at the 63× objective with immersion oil as previously described (43).

## Electron microscopy

A 100 µL volume of inoculated culture broth was mixed with 100 µL of 4× glutaraldehyde and then stirred for 30–60 minutes at room temperature. A 20 µL volume of 10% phosphotungstic acid was added and stirred again for 5 minutes, deposited on a Cytospin (Thermo Fisher Scientific, Waltham, MA, USA), and centrifugated for 7 minutes at 800 rpm. Slides were read with the Scanning Electron Microscope TM4000 plus (Hitachi, Tokyo, Japan). Micrographs were acquired at high magnifications ranging from 2,500× to 10,000× with an accelerating voltage of 10 kV using the Backscatter Electron detector under the high vacuum mode.

## Whole genome sequencing

DNA was extracted from 200 µL of culture as previously described (45) using the EZ1 DNA Tissue Kit (Qiagen, Courtaboeuf, France) after overnight incubation at 56°C with 20 µL proteinase K (Qiagen). Glass powder was added to the mix, incubated for 20 minutes at 100°C and immediately vortexed 90 s at 6.5 m/s with the FastPrep instrument (MP Biomedical Europe, Illkirch, France). The mix was centrifuged for 5 minutes at $17,000 \times g$ and DNA was extracted from 200 µL of supernatant and eluted in a 50 µL volume. The methanogen identification was first done by partial 16S rRNA gene PCR-amplification followed by Sanger sequencing (primers: SDArch0333aS15, 5′-TCCA GGCCCTACGGG-3′ and SDArch0958aA19, 5′-YCCGGCGTTGAMTCCAATT-3′) as previously described (46). Generated reads were assembled by ChromasPro software (version 1.34), and then aligned to the NCBI GenBank database using the Blast platform. DNA was then engaged for WGS using the MiSeq Illumina pair-end protocol (Illumina, San Diego, CA, USA) and Oxford Nanopore single-long reads (Oxford Nanopore Technologies, Oxford, UK) platforms as previously described (47, 48). Quality control of NGS data was done using the fastQC command on the Galaxy Europe online platform (https://usegalaxy.eu/). Illumina and Nanopore reads were concatenated and *de novo* assembled using Spades software version (version 3.15.4) and generated contigs were blasted against the NCBI GenBank database. The quality of genome assemblies was checked using CheckM (49). The ANI analysis was performed with the 38 genomes included in this study (Table S3) using PYANI software version (0.2.7) with standard parameters, and genome sequences with >95% identity corresponding to the same species (50). dDDH (DNA-DNA hybridization) values were calculated using the TYGS [Type (Strain) Genome Server] online tool (51) based on the 38 included genome sequences. The threshold for DDH values indicating membership in the same species was above 70% (52). Phylogenetic analysis based on WGS-derived 16S rRNA gene sequences with the 100 hit-blasts was performed using Neighbor-Join and BioNJ algorithms standard parameters on MEGA software (version 7.0.26). The evolutionary history was inferred using the Maximum Likelihood method based on the JTT matrix-based model and 1,000 replicates bootstrap consensus. Branches corresponding to partitions reproduced in less than 90% of bootstrap replicates are collapsed. The initial tree was automatically obtained by applying Neighbor-Join and BioNJ algorithms to a matrix of pairwise distances estimated

using a JTT model, then selecting the topology with the superior log likelihood value. All positions containing gaps and missing data were eliminated, and a total of 1138 positions were included in the final data set (Fig. 3). Genome representation was performed using the Proksee platform with standard parameters (version 1.0.0; https://proksee.ca/). The genome was annotated on the DDBJ Fast Annotation and Submission Tool online platform (https://dfast.ddbj.nig.ac.jp/) and putative enzymes involved in the utilization of different substrates for methanogenesis were sought. Phylogenetic trees of NADPH-dependent butanol dehydrogenase and aldehyde dehydrogenase gene sequences and protein sequences were obtained using Neighbor-Join and BioNJ algorithms standard parameters on MEGA software (version 7.0.26). The evolutionary history was inferred using the maximum likelihood method based on the JTT matrix-based model (Figures S3a through d). Blastp was used against the COG database to find putative encoded protein functions for *Ca*. M. massiliense and other methanogens (Table S1).

## MALDI-TOF mass spectrometry

Protein extraction was performed as previously described (53) from 1 mL of pure liquid cultures of the new isolate and *M. stadtmanae* DSMZ3091 (CSUR P9634). Two deposits were made for each sample and peptide profiles were determined using an Autoflex II mass spectrometer equipped with a 337 nm nitrogen laser (Brüker Daltonics, Bremen, Germany). Protein extract of *E. coli* DH5α (Brüker Daltonics) was used as a positive control and non-inoculated medium as a negative control. Spectra were recorded in the positive linear mode for masses ranging from 2 to 20 kDa (parameter settings: ion source 1 [IS1], 20 kV; IS2, 18.5 kV; lens, 7 kV). A spectrum was obtained after 675 shots with a variable laser power. The software tool AutoXecute acquisition control was applied for the automated data acquisition. Peptidic spectra analysis and comparison were performed using FlexAnalysis 2.4 software (Brüker Daltonics).

## Antibiotic susceptibility testing

Antibiotic susceptibility was tested using a previously described macrodilution technique and based on *M. stadtmanae* susceptibility (54). Amoxicillin, imipenem, metronidazole, bacitracin, fusidic acid, and chloramphenicol (Sigma Aldrich, Saint Quentin Fallavier, France) were dissolved in sterile water and added separately to the broth medium to obtain a final concentration of 100 mg/L for amoxicillin and imipenem, 1 mg/L for metronidazole, 4 mg/L for chloramphenicol and 4 mg/L for bacitracin and fusidic acid. For each antibiotic, two Hungate tubes containing 4.5 mL of liquid medium were inoculated with $10^5$ colony-forming units (CFU)/mL of methanogen, and one control tube was inoculated with 500 µL of PBS. Two Hungate tubes containing 4.5 mL of liquid medium without antibiotics inoculated with $10^5$ CFU/mL of methanogen were used as positive controls. Culture tubes were incubated for 5 days at 37°C in an anaerobic atmosphere and were monitored using gas chromatography at day 0 (D0) and day 5 (D5). 100 µL was sampled from the headspace of the tube and injected into a Clarus 580 chromatograph (Perkin Elmer, Waltham, MA, USA). The so-called "sensitive" method was used and the resulting area under the curve values and their conversion to molar concentration (mol/L) were used to quantify methane production. The minimum inhibitory concentration (MIC) was defined as the lowest antibiotic concentration that inhibited methane production. The following controls were performed for each antibiotic to verify its activity: *E. coli* CSUR (Collection Souches Unité des Rickettsies, Marseille, France) Q6942 and *S. aureus* CSUR Q7280 whose antibiotic susceptibility was previously determined, were each seeded on two Columbia agar 5% sheep blood (COS) media (bioMérieux, Marcy-Étoile, France) and two additional COS media were inoculated with the same antibiotic solution used for each experiment, two non-seeded COS media were inoculated with the antibiotic solution and two others with PBS.

## pH susceptibility testing

Five pH conditions were tested after the pH of the liquid culture medium was adjusted to pH 2, 4, 6, 8, and 10 using HCl or KOH (Sigma Aldrich, Saint Quentin Fallavier, France). For each condition, two Hungate tubes containing 4.5 mL of liquid culture medium were inoculated with $10^5$ CFU/mL of the methanogen, and one Hungate tube was inoculated with PBS as a negative control. Positive controls consisted of two Hungate tubes containing liquid medium at pH 7.3 inoculated with $10^5$ CFU/mL. The culture tubes were incubated for 7 days at 37°C in an anaerobic atmosphere and were monitored using gas chromatography at day 0 (D0) and day 7 (D7) as described above. The methanogen was considered susceptible to a pH value when no methane was detected after a 7-day incubation. Methane-positive cultures were then subcultured in GG medium at pH 7.3 and 37°C for 7 days to ensure the viability of the methanogen.

## Substrate affinity testing

A basal GG medium (GG0) derived from the original GG medium but without acetate, formate, methanol or ethanol was used, and each substrate or substrate combinations were added separately in the medium GG0. Nine conditions were tested using GG0 medium, six under nitrogen atmosphere: no substrate, acetate (2 g/L), formate (2 g/L), methanol (2% vol/vol), ethanol (2% [vol/vol]), methanol and ethanol (2% [vol/vol]), and three under 5% $H_2$/20% $CO_2$/75% $N_2$ atmosphere without any substrate, with methanol or with ethanol. For each condition, three Hungate tubes containing 4.5 mL of liquid culture medium were inoculated with $10^5$ CFU/mL of the methanogen, and one Hungate tube was inoculated with PBS as a negative control. The culture tubes were incubated for 7 days at 37°C and were monitored using gas chromatography at day 0 (D0) and day 7 (D7) as described above.

## Human and animal faeces molecular screening

The prevalence of the methanogen was assessed in 150 human and 313 animal stool samples using a real-time (RT) PCR assay, with the fecal samples coming from 52 dogs, 20 sheep, 75 horses, 52 cows, 50 red kangaroos, and 64 pigs (Fig. S4). DNA was extracted as described above from 0.5 g put in 500 µL of G2 buffer (Qiagen, Courtaboeuf, France). We used phylogeny and annotation of the methanogen genome sequence to investigate and identify a candidate sequence for RT-PCR screening of a large stool panel from humans and animals. Because the 16S rRNA-encoding gene was not informative for the screening, the DNA polymerase encoding gene was targeted for primer design. Using the Primer3 program (https://primer3.org/), an in-house RT-PCR system was designed, targeting 124 bp of *Ca.* M. massiliense polymerase encoding gene including a 6-carboxyfluorescein [FAM]- 5'-AGCACGTATGACTACAGGACA-3' probe, forward primer (5'-AACACCGCTTACTGTTGCAC-3') and reverse primer (5'-ACTCGTCCGTAT CTGGCTTTA-3'). RT-PCR specificity was *in silico* tested by the NCBI BLAST program (http://www.ncbi.nlm.nih.gov/BLAST) using each primer alone and *in-silico* amplified using both primers and the specific probe, particularly against *Methanosphaera* sp. Mb5, RUG761, SHI1033, WGK6 genome sequences, using Amplify4 software (version 1.0, Bill Engels, University of Wisconsin). The RT-PCR amplification program was 95°C for 15 minutes, followed by 39 cycles of 95°C for 30 s and 60°C for 1 minutes, and a final step of 40°C for 30 s. RT-PCR validation was done in a 20 µL volume containing 5 µL DNA, 20 µM of each primer, and 10 µM of the specific probe, using the Roche Master Mix according to the manufacturer's protocol (Roche, Indianapolis, IN, USA). To disclose any cross-contamination, 5 µL of DNase/RNase-free distilled water was used as a negative control. The specificity was tested with *M. smithii* Q5488, *M. oralis* Q6268 preserved in the CSUR, and *M. stadtmanane* DSM3091 (CSUR P9634). As a supplementary control, the methanogens 16S rRNA gene was also amplified as previously described to ensure the presence of methanogens (38). A calibration curve was obtained using serial dilution

from $10^8$ CFU/mL (0.5 McFarland turbidity standard) to $10^1$ CFU/mL and was used to quantify the load of *Ca.* M. massiliense in each sample.

## Amplicon sequencing

The amplicons obtained were sequenced according to Sangers' method using the BigDye Terminator cycle sequencing kit DNA (Applied Biosystems, Foster City, USA) with the forward primer (5′-AACACCGCTTACTGTTGCAC-3′) and reverse primer (5′-ACTCGTCCG TATCTGGCTTTA-3′) from our RT-PCR system described above. Sequences were assembled with Chromas Pro software, version 1.7 (Technelysium Pty Ltd., Tewantin, Australia) and compared with sequences available in the GenBank database using the online NCBI BLAST program (https://blast.ncbi.nlm.nih.gov/Blast.cgi).

## ACKNOWLEDGMENTS

This work was supported by Institut Hospitalo-Universitaire (IHU) Méditerranée Infection (Marseille, France). V.P. benefited from a PhD grant from Aix-Marseille Université.

The authors thank the Hitachi team (Japan) for the collaborative study with Hitachi High Tech Corporation and Institut Hospitalo-Universitaire (IHU) Méditerranée Infection and the installation of the TM4000 Plus microscope in our facility.

## AUTHOR AFFILIATIONS

[1]Aix-Marseille Université, IRD, MEPHI, IHU Méditerranée Infection, Marseille, France
[2]Aix-Marseille Université, Ecole de Médecine Dentaire, Marseille, France
[3]IHU Méditerranée Infection, Marseille, France
[4]Veterinary Service, Beauval Zoopark, Saint-Aignan, France

## PRESENT ADDRESS

Madjid Morsli, VBIC, INSERM U1047, Montpellier Université, Service de Microbiologie et Hygiène Hospitalière, CHU Nîmes, Nîmes, France
Cheick Oumar Guindo, Sorbonne Université, INSERM, CNRS, Centre d'Immunologie et des Maladies Infectieuses (CIMI), Paris, France

## AUTHOR ORCIDs

Michel Drancourt  http://orcid.org/0000-0003-0768-1139
Ghiles Grine  http://orcid.org/0000-0002-5271-3307
Elodie Terrer  http://orcid.org/0000-0002-5788-552X

## FUNDING

| Funder | Grant(s) | Author(s) |
| --- | --- | --- |
| Aix-Marseille Université (AMU) | | Cheick Oumar Guindo |
| | | Bernard Davoust |
| | | Michel Drancourt |
| | | Gérard Aboudharam |
| | | Elodie Terrer |
| | | Virginie Pilliol |
| | | Madjid Morsli |
| | | Laureline Terlier |
| | | Yasmine Hassani |
| | | Ihab Malat |

This work received funding from the Twinning European project: 952583— MICAfrica—H2020-WIDESPREAD-2018-2020/H2020-WIDESPREAD-2020-5

## AUTHOR CONTRIBUTIONS

Virginie Pilliol, Conceptualization, Data curation, Formal analysis, Investigation, Methodology, Software, Visualization, Writing – original draft | Madjid Morsli, Investigation, Software | Laureline Terlier, Investigation | Yasmine Hassani, Investigation | Ihab Malat, Methodology | Cheick Oumar Guindo, Investigation | Bernard Davoust, Investigation | Benjamin Lamglait, Investigation | Michel Drancourt, Conceptualization, Supervision, Validation, Writing – original draft, Writing – review and editing | Gérard Aboudharam, Funding acquisition, Supervision, Writing – review and editing | Ghiles Grine, Conceptualization, Methodology, Supervision, Validation, Writing – original draft, Writing – review and editing | Elodie Terrer, Conceptualization, Formal analysis, Funding acquisition, Methodology, Project administration, Resources, Supervision, Validation, Visualization, Writing – review and editing

## DATA AVAILABILITY

All genomic data were submitted to the GenBank NCBI database under accession number OP930955 (16S rRNA sequence) and JARBXM000000000 (genome). The amplicon sequences were also deposited under accession numbers: OX456088; OX456151; OX456152; OX456153; OX456154; OX456155; OX456156; OX456157; OX456158; OX456157; OX456158; OX456159; OX456160; OX456161; OX456162; OX456163; OX456164; OX456165; OX456166; OX456167; OX456168; OX456169; OX456170.

## ADDITIONAL FILES

The following material is available online.

### Supplemental Material

**Figure S1 (Spectrum05141-22-s0001.jpg).** Hydrogen-free and carbon dioxide-free culture protocol in GG medium for isolation of *Candidatus* Methanosphaera massiliense sp. nov.

**Figure S2 (Spectrum05141-22-s0002.jpg).** *M. stadtmanae* DSM3091 and *Candidatus* Methanosphaera massiliense sp. nov. MALDI-TOF mass spectrometry spectra.

**Figure S3 (Spectrum05141-22-s0003.pdf).** Phylogenetic tree based on NADPH-dependent butanol dehydrogenase protein and nucleotide sequence and aldehyde dehydrogenase analysis protein and nucleotide sequence.

**Figure S4 (Spectrum05141-22-s0004.jpg).** Map of France locating the sources of animal and human fecal samples investigated.

**Table S1 (Spectrum05141-22-s0005.xlsx).** Clusters of Orthologous Groups (COGs) functional categories assignation and representation of *Ca*. M. massiliense and other methanogens.

**Table S2 (Spectrum05141-22-s0006.xlsx).** *Candidatus* Methanosphaera massiliense sp. nov. and the first 100 hit blasts downloaded from NCBI GenBank Database.

**Table S3 (Spectrum05141-22-s0007.xlsx).** Methanogen genomes used in this study.

**Table S4 (Spectrum05141-22-s0008.xlsx).** Data for pH susceptibility testing, antibiotic susceptibility testing, and substrate affinity testing.

**Table S5 (Spectrum05141-22-s0009.xlsx).** Testing specificity of the designed RT-PCR system for *Ca*. M. massiliense sp. nov.

**Table S6 (Spectrum05141-22-s0010.xlsx).** Prevalence of *Ca*. M. massiliense in human and animal faecal samples and amplicon sequencing.

### Open Peer Review

**PEER REVIEW HISTORY (review-history.pdf).** An accounting of the reviewer comments and feedback.

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
