## [Reviewer comments · Microbiology Spectrum]

Microbiology Spectrum

***Candidatus* Methanosphaera massiliense sp. nov., a methanogenic archaeal species found in a human fecal sample and being prevalent in pigs and red kangaroos.**

Virginie Pilliol, Madjid Morsli, Laureline Terlier, Yasmine Hassani, Ihab Malat, Cheick GUINDO, Bernard DAVOUST, Benjamin Lamglait, Michel Drancourt, Gérard Aboudharam, Ghiles Grine, and Elodie Terrer

Corresponding Author(s): Elodie Terrer, Aix-Marseille Universite

Review Timeline:

Submission Date:	December 14, 2022
Editorial Decision:	January 30, 2023
Revision Received:	April 24, 2023
Editorial Decision:	May 22, 2023
Revision Received:	October 6, 2023
Editorial Decision:	October 30, 2023
Revision Received:	November 20, 2023
Accepted:	November 24, 2023

Editor: Henning Seedorf

Reviewer(s): The reviewers have opted to remain anonymous.

Transaction Report:

DOI: <https://doi.org/10.1128/spectrum.05141-22>

January 30, 2023

Dr. Elodie Terror
Aix-Marseille Universite
MARSEILLE
France

Re: Spectrum05141-22 (Methanosphaera massiliense sp. nov. questioned zoonotic sources of human-associated methanogens.)

Dear Dr. Elodie Terror:

Link Not Available

Sincerely,

Henning Seedorf

Journals Department
Reviewer comments:

Reviewer #2 (Public repository details (Required)):

Raw sequence data and assembly genome each need to be deposited in NCBI. Currently only the partial 16S sequence has been deposited and accession cited in text.

Reviewer #2 (Comments for the Author):

Authors have isolated a new species of Methanosphaera and proposed the species designation massiliense BB6. PCR screening of a human cohort and animal samples would suggest this is not a species commonly found in the human gut and is likely a zoonotic transfer. It would have been of interest to follow the 1 participant (out of 150) who was positive for BB6 to see if

the strain persisted or was transient due to livestock exposure. The paper would benefit from some additional work, namely culture-based work to confirm the substrate affinity of the strain and its affinity for each of the methanogenic pathways. The introduction would benefit from more specific details and additional references.

Specific Comments

1. Line 55: I think authors should specify this study was only conducted in children and authors should also keep in mind that of the two groups reported in this study only 40% of controls were positive for *M. smithii* (not 100%). I would suggest caution here in the introduction to not mislead readers into thinking an absence or reduced abundance of methanogenic archaea is automatically associated with dysbiosis.
2. Line 57: there are more recent works that can be included in citation here to demonstrate the work done to recover *Methanosphaera* spp. from the human GIT. One example being <https://www.ncbi.nlm.nih.gov/pmc/articles/PMC6246560/>
3. Line 58-59: I do not think authors have interpreted this reference correctly. If there are other species of *Methanosphaera* implicated with human disease they are not described in this min-review. Please instead include the relevant research paper to support this background material. I do not think these references support the link to human disease (few studies currently achieve this). I think it would be more appropriate to present the knowledge existing around the differential inflammatory potential of *Methanosphaera* versus other genus and then suggest that *Methanosphaera* have a role to play in human disease.
4. Line 60-63: if authors only want to focus on isolated representatives this list is appropriate. If authors would like to focus on isolates and metagenome strains sheep can also be included in this list. If authors want to include animals that have 16S representative *Methanosphaera* sequences this list is much more extensive and can be found in their ref 15. Please clarify here.
5. Line 64: Authors need to provide more background information and appropriate references to convince readers that there are species other than *stadtmanae* in the human gut. All current literature (metagenome recovery and 16S profiling) that I am aware of would suggest that *stadtmanae* held the monopoly.
6. Line 73: It is not clear in the methods what primers were used to initially screen stool samples.
7. Line 78: *Methanosphaera* spp. are Gram-positive, are authors sure that the Gram-variable cells were not a contaminant?
8. Line 82: Following comment above are authors sure that the MALDI-TOF returned a nil result due to a database fail rather than the culture not being pure?
9. Line 85: (inc methods section) why did authors not use antibiotic strip testing for this assay? Authors report a lawn forming on agar so this should be possible and should yield a more accurate MIC?
10. Line 86: rationale for these antibiotics?
11. Line 85: provide data/results/plots from "sensitive" method testing (AUC from GC-MS data) as supplementary
12. Line 115: can authors please also include CheckM stats for BB6 (genome completeness and contamination percentages).
13. Line 133: can authors describe origin and choice of *massiliense* as the species name in text here.
14. Line 134: instead of referring to the sample as leftover stool please be more precise and describe the host e.g., "healthy xx-year-old male".
15. Line 135: if the strain has been publicly deposited, please describe this in text. If the strain is only held by the authors, then this is the 3rd *Methanosphaera* spp. to be isolated from humans (DSMZ3091, PA5 and BB6).
16. Line 137: did authors generate the peptide fingerprint for *stadtmanae*? It would be worthwhile having this represented beside BB6 as a panelled figure for comparison.
17. Line 139: For me this not a very thorough description of the new species and is a basic in the sense that authors have not completed any substrate affinity assays to confirm the growth/methane production capabilities of BB6. Neither have the investigated the genome to report the molecular potential of BB6 to utilise different methanogenic pathways.
18. Line 143-152: this whole section belongs in the results and in some respects addresses my point above. As authors have BB6 in culture it is a simple task to complete a growth rate under each methanogenic substrate combination to validate findings seen in the genome.
19. Line 161: what was the occupation of the donor? Was it likely they would have had livestock exposure? What do authors mean by pig-derived foodstuffs? Are they referring to meat? I highly doubt methanogenic archaea (an oxygen sensitive microbe) would survive this amount of handling before ingestion.
20. Line 177: humans are mammals, please rephrase this.
21. Line 184: please provide a description of the donor the isolate was generated from, at a minimum age, sex, health status (and ethnicity).
22. Line 200: as the previous publication has not been released please describe here how the SAB media was modified to create GG.
23. Line 206: what substrate sources were included? And what was the concentration?
24. Line 294: what Illumina platform was used?
25. Line 321-329: please describe qPCR cycle conditions here.
26. Line 340: this data deposit is only for the partial 16S gene. Please include the other accession IDs or if not completed yet please also deposit the raw sequence reads and WGS to NCBI and provide these accession IDs.
27. Figures: resolution of most images are very poor and unsuitable for publication, figure legends also need to be clearer and figures which are panels need to be formatted accordingly before upload.
28. Figure 1: significantly reduce white space in this image. I would even recommend moving this to supplementary material as it contains only generic images and no results generated by the authors.
29. Figure 2: this image needs to be panelled and labelled with A-C before reupload. Resolution needs to be improved. Scale bar needs to be included on 2A and cropped to include only those cells in focus. The scale of the image also makes it near impossible to see the cells without having to zoom in. 2B can also be cropped to remove dead space. 2C comes up as 3 images on my manuscript copy and is poor quality. Is this image to show the autofluorescence? If so a better quality image needs to be

taken before resubmission. Figure legend can also be more descriptive.

30. Figure 3: legend needs to be more descriptive so anyone not familiar with MALDI-TOF output can interpret the results. I would also recommend panelling this with a comparison of the DSMZ3091 profile.

31. Figure 5: line 477-483: this a mixture of methods and results. Please condense this information in the figure legend and provide in the results and methods section where appropriate.

Reviewer #3 (Public repository details (Required)):

The genome is not provided.

Reviewer #3 (Comments for the Author):

General remarks

- The article does not represent a valid description of the new culture. Therefore, the isolate should be referred to as "Candidatus *M. massiliense*" throughout.
- In the spirit of Open Science, please provide the accession number for the genome in question, not just for the 16S rRNA gene.
- The new isolate should be included in a public culture collection and made available to all researchers.
- The genome sequence was not compared to the largest source of data for archaeal genomes from the human gut (Chibani et al., 2022); this needs to be included to put the result in context.
- The zoonotic prevalence of *Cand. M. massiliense* is a very strong statement and requires additional confirmation through appropriate assessment of the approach (i.e., sequencing of RT-PCR products or metagenomics) and inclusion of ct values for all samples and controls and calculation of copy numbers.

Specific remarks

- Line 13: running head is not appropriate, zoonotic origin has not been shown.
- Please check the title, it does not make sense and is probably incorrect (questions?)
- Line 24: This is not correct. Next to *M. stadtmannae*, also other methanospaera representatives have been detected via molecular methods, e.g. by genome binning. Please check Chibani et al. 2022.
- Line 27 here and everywhere: should be *Candidatus M. massiliense*
- Line 30: rephrase, a gap is not expected in a microbial genome? You mean you were able to produce a gapless genomic sequence, correct?
- Line 33 novel screening systems require proper evaluation, e.g. sequencing of the qPCR results, otherwise these statements cannot be made.
- Line 36: Why does your finding question the possible zoonotic source? It rather opens the questions on the origin... Maybe rephrase, the meaning is not clear.
- Line 39: the only cultivated *Methanospaera* representative
- Line 42: This is a strong claim and has to be supported by e.g. proper validation of the qPCR primers by sequencing, or metagenomic sequencing of the samples used.
- Line 50 ff: also other members have been isolated, namely *Cand. M. intestini*, *Cand. M. intestinalis*, *Cand. M. alvus*. Please update your references.
- Line 69: it is not clear whether it is novel per se, as its sequence was not put in context with retrieved genomic bins from human gut (again, see Chibani et al).
- Line 74/133: consider removing "leftover" here, as this is confusing for the reader. It is explained in the materials and methods.
- Line 80: showed auto-fluorescence
- Line 81: Was the purity of the final culture checked, and how?
- Line 97: 20 different *Methanospaera* genomes representing unique strains were detected in Chibani et al., please put your genome in context (ANI differences, tree)
- Please give further information on the genomic sequence, e.g. the completeness of marker genes, and contamination (e.g. use CheckM)
- Line 106: cover = coverage?
- Line 114: *Candidatus*
- Line 114: please provide genome sequence in a public repository; also the culture should be placed in a public collection
- Line 118: evaluation of the new qPCR primers and set-up has to be done more thoroughly.
- Line 122: please report quantitative data, not only qualitative information (presence vs. non-present). It is important to know the relative abundance, not only the prevalence.
- Line 129: what is the purpose of this Nanoarchaea research chapter?
- Line 135: please give details: to which public collection was it submitted? And, it is not thoroughly described: no proper description is made. <https://www.microbiologyresearch.org/content/journal/ijsem?page=about-journal#2>
- Line 136: do you mean light and electron microscopy?
- Line 138: this does not mean it is properly evaluated
- Line 146ff: Please provide the genome sequence in a public database. Until this is not done, such statements are not useful to

any reader

- Line 157: did you really check whether it remained alive? Or was this the range of cultivation?
- Line 200: The composition of the medium is crucial to this publication. Either, include a citation or describe the medium.
- Line 210: how exactly was the purity ensured?
- Line 341: the accession number does not refer to the submission of the genome. This only includes the 16S rRNA gene.
- Table 1: Is the genome closed or not? Indicate contamination and completeness of the genome
- Fig.1: explain what is shown on the figure. What is the picture on the right?
- Fig.2: arrange all pictures for Fig. 2 in one panel
- Fig. 2a should be cropped and has to include a bar.
- Same applies to Fig. 2b, provide more details on the technique itself (negative stain, shadowed... etc.).
- Move Fig. 3 to supplementary
- Fig. 4: add more explanation to the figure itself (legend?)
- Fig. 7: what is an atomic representation?

Staff Comments:

Preparing Revision Guidelines

Please return the manuscript within 60 days; if you cannot complete the modification within this time period, please contact me. If you do not wish to modify the manuscript and prefer to submit it to another journal, please notify me of your decision immediately so that the manuscript may be formally withdrawn from consideration by Microbiology Spectrum.

Methanosphaera massiliense sp. nov. questioned zoonotic sources of human associated methanogens.

2023 PILLIOL

Authors have isolated a new species of Methanosphaera and proposed the species designation massiliense BB6. PCR screening of a human cohort and animal samples would suggest this is not a species commonly found in the human gut and is likely a zoonotic transfer. It would have been of interest to follow the 1 participant (out of 150) who was positive for BB6 to see if the strain persisted or was transient due to livestock exposure. The paper would benefit from some additional work, namely culture-based work to confirm the substrate affinity of the strain and its affinity for each of the methanogenic pathways. The introduction would benefit from more specific details and additional references.

Specific Comments

1. Line 55: I think authors should specify this study was only conducted in children and authors should also keep in mind that of the two groups reported in this study only 40% of controls were positive for *M. smithii* (not 100%). I would suggest caution here in the introduction to not mislead readers into thinking an absence or reduced abundance of methanogenic archaea is automatically associated with dysbiosis.
2. Line 57: there are more recent works that can be included in citation here to demonstrate the work done to recover *Methanosphaera* spp. from the human GIT. One example being <https://www.ncbi.nlm.nih.gov/pmc/articles/PMC6246560/>
3. Line 58-59: I do not think authors have interpreted this reference correctly. If there are other species of *Methanosphaera* implicated with human disease they are not described in this min-review. Please instead include the relevant research paper to support this background material. I do not think these references support the link to human disease (few studies currently achieve this). I think it would be more appropriate to present the knowledge existing around the differential inflammatory potential of *Methanosphaera* versus other genus and then suggest that *Methanosphaera* have a role to play in human disease.
4. Line 60-63: if authors only want to focus on isolated representatives this list is appropriate. If authors would like to focus on isolates and metagenome strains sheep can also be included in this list. If authors want to include animals that have 16S representative *Methanosphaera* sequences this list is much more extensive and can be found in their ref 15. Please clarify here.
5. Line 64: Authors need to provide more background information and appropriate references to convince readers that there are species other than *stadtmanae* in the human gut. All current literature (metagenome recovery and 16S profiling) that I am aware of would suggest that *stadtmanae* held the monopoly.
6. Line 73: It is not clear in the methods what primers were used to initially screen stool samples.
7. Line 78: *Methanosphaera* spp. are Gram-positive, are authors sure that the Gram-variable cells were not a contaminant?
8. Line 82: Following comment above are authors sure that the MALDI-TOF returned a nil result due to a database fail rather than the culture not being pure?

9. Line 85: (inc methods section) why did authors not use antibiotic strip testing for this assay? Authors report a lawn forming on agar so this should be possible and should yield a more accurate MIC?
10. Line 86: rationale for these antibiotics?
11. Line 85: provide data/results/plots from "sensitive" method testing (AUC from GC-MS data) as supplementary
12. Line 115: can authors please also include CheckM stats for BB6 (genome completeness and contamination percentages).
13. Line 133: can authors describe origin and choice of massiliense as the species name in text here.
14. Line 134: instead of referring to the sample as leftover stool please be more precise and describe the host e.g., "healthy xx-year-old male".
15. Line 135: if the strain has been publicly deposited, please describe this in text. If the strain is only held by the authors, then this is the 3rd Methanosphaera spp. to be isolated from humans (DSMZ3091, PA5 and BB6).
16. Line 137: did authors generate the peptide fingerprint for stadtmannae? It would be worthwhile having this represented beside BB6 as a panelled figure for comparison.
17. Line 139: For me this not a very thorough description of the new species and is a basic in the sense that authors have not completed any substrate affinity assays to confirm the growth/methane production capabilities of BB6. Neither have the investigated the genome to report the molecular potential of BB6 to utilise different methanogenic pathways.
18. Line 143-152: this whole section belongs in the results and in some respects addresses my point above. As authors have BB6 in culture it is a simple task to complete a growth rate under each methanogenic substrate combination to validate findings seen in the genome.
19. Line 161: what was the occupation of the donor? Was it likely they would have had livestock exposure? What do authors mean by pig-derived foodstuffs? Are they referring to meat? I highly doubt methanogenic archaea (an oxygen sensitive microbe) would survive this amount of handling before ingestion.
20. Line 177: humans are mammals, please rephrase this.
21. Line 184: please provide a description of the donor the isolate was generated from, at a minimum age, sex, health status (and ethnicity).
22. Line 200: as the previous publication has not been released please describe here how the SAB media was modified to create GG.
23. Line 206: what substrate sources were included? And what was the concentration?
24. Line 294: what Illumina platform was used?
25. Line 321-329: please describe qPCR cycle conditions here.
26. Line 340: this data deposit is only for the partial 16S gene. Please include the other accession IDs or if not completed yet please also deposit the raw sequence reads and WGS to NCBI and provide these accession IDs.
27. Figures: resolution of most images are very poor and unsuitable for publication, figure legends also need to be clearer and figures which are panels need to be formatted accordingly before upload.
28. Figure 1: significantly reduce white space in this image. I would even recommend moving this to supplementary material as it contains only generic images and no results generated by the authors.
29. Figure 2: this image needs to be panelled and labelled with A-C before reupload. Resolution needs to be improved. Scale bar needs to be included on 2A and cropped to include only those cells in focus. The scale of the image also makes it near impossible to see the cells

without having to zoom in. 2B can also be cropped to remove dead space. 2C comes up as 3 images on my manuscript copy and is poor quality. Is this image to show the autofluorescence? If so a better quality image needs to be taken before resubmission. Figure legend can also be more descriptive.

30. Figure 3: legend needs to be more descriptive so anyone not familiar with MALDI-TOF output can interpret the results. I would also recommend panelling this with a comparison of the DSMZ3091 profile.
31. Figure 5: line 477-483: this a mixture of methods and results. Please condense this information in the figure legend and provide in the results and methods section where appropriate.

Reviewer #2 (Comments for the Author):

Authors have isolated a new species of *Methanosphaera* and proposed the species designation *massiliense* BB6. PCR screening of a human cohort and animal samples would suggest this is not a species commonly found in the human gut and is likely a zoonotic transfer. It would have been of interest to follow the 1 participant (out of 150) who was positive for BB6 to see if the strain persisted or was transient due to livestock exposure. The paper would benefit from some additional work, namely culture-based work to confirm the substrate affinity of the strain and its affinity for each of the methanogenic pathways. The introduction would benefit from more specific details and additional references.

The authors would like to thank the reviewer for his comments. Although we cannot provide further information about the donor for ethical reasons, we hope that this new manuscript will give complete satisfaction to the reviewer, in particular on the substrate affinity of *Ca. M. massiliense*.

Specific Comments

1. Line 55: I think authors should specify this study was only conducted in children and authors should also keep in mind that of the two groups reported in this study only 40% of controls were positive for *M. smithii* (not 100%). I would suggest caution here in the introduction to not mislead readers into thinking an absence or reduced abundance of methanogenic archaea is automatically associated with dysbiosis.

The sentence was modified as follow :

« An exception is the dysbiotic depletion of *M. smithii* and its absence in children with deadly severe acute malnutrition (8). » **(Lines 57-59)**

2. Line 57: there are more recent works that can be included in citation here to demonstrate the work done to recover *Methanosphaera* spp. from the human GIT. One example being <https://www.ncbi.nlm.nih.gov/pmc/articles/PMC6246560/>

The whole paragraph was modified as follow, including corrections asked by the other reviewer :

“Whereas *M. stadtmanae* was for a long time the sole *Methanosphaera* member cultured from human faeces (5) and further detected by PCR-based methods in human dental plaque (25), a new human isolate, *Methanosphaera* sp. PA5 was provided by Hoedt *et al.* in 2018 (26). Furthermore, *Methanosphaera* have been cultured from mammal guts including *Methanosphaera cuniculi* (*M. cuniculi*) in rabbit (27), *Methanosphaera* sp. WGK6 in Western grey kangaroo (*Macropus fuliginosus*) (28) and *Methanosphaera* sp. BMS in cow guts (26). Recently, Hoedt *et al.* (26) and Chibani *et al.* (9) obtained respectively 7 and 20 *Methanosphaera* spp. genomes from metagenomic data of mammals (including humans) gut as reported in **supplementary data 4**, some of them not corresponding to known isolates. Particularly, among the 20 *Methanosphaera* spp. genomes provided by Chibani *et al.* from the human gut, 17 were related to *M. stadtmanae*, two to *M. cuniculi* and one to a genome obtained from cow rumen, suggesting possible host transfers and adaptation from animals to human gut. **(Lines 79-92)**

3. Line 58-59: I do not think authors have interpreted this reference correctly. If there are

other species of *Methanosphaera* implicated with human disease they are not described in this min-review. Please instead include the relevant research paper to support this background material. I do not think these references support the link to human disease (few studies currently achieve this). I think it would be more appropriate to present the knowledge existing around the differential inflammatory potential of *Methanosphaera* versus other genus and then suggest that *Methanosphaera* have a role to play in human disease.

The whole paragraph was replaced as follow :

“Since the detection of *Methanosphaera* spp. and *Methanobrevibacter* spp. in the bioaerosols from poultries, dairy farms and piggeries (14–16), studies were conducted in animal models to investigate the role of methanogens in hypersensitivity diseases. The administration of *M. stadtmanae* extract aerosol but not *M. smithii* aerosol in the airways of mice yielded to a hypersensitivity response with an accumulation of eosinophils and neutrophils in the lungs (17, 18), laying the groundwork for the potentially pathogenic role of *M. stadtmanae*. The pro-inflammatory capacity of *M. stadtmanae* was next used to induce hypersensitivity pneumonitis disease in mice models (19, 20). Moreover, further studies focused on human cells response exposition to methanogens, *M. smithii* and *M. stadtmanae* are both able to activate mononuclear cells but *M. stadtmanae* induced a higher release of pro-inflammatory cytokines than *M. smithii* (21, 22). *M. stadtmanae* phagocytosis is crucial for cell activation (21). At the molecular level, *M. stadtmanae* RNA was identified as the microbe-associated molecular pattern (MAMP) that can trigger a NLRP3 inflammasome activation through the toll-like receptor 8 (TLR8) (23). Interestingly, *M. stadtmanae* was more prevalent in patients with inflammatory bowel syndrome which developed a specific IgG response to this methanogen (22). More recently, a decrease in the *Methanosphaera* abundance was noticed from healthy to long-time diabetic patients whereas an increase of *M. smithii* was observed (24).” (Lines 60 to 78)

4. Line 60-63: if authors only want to focus on isolated representatives this list is appropriate. If authors would like to focus on isolates and metagenome strains sheep can also be included in this list. If authors want to include animals that have 16S representative *Methanosphaera* sequences this list is much more extensive and can be found in their ref 15. Please clarify here.

We would like to focus on isolated *Methanosphaera* species but we also added the metagenomes provided by Hoedt *et al.* (2018) and Chibani *et al.* (2022). Please see lines 79 to 92.

5. Line 64: Authors need to provide more background information and appropriate references to convince readers that there are species other than *stadtmanae* in the human gut. All current literature (metagenome recovery and 16S profiling) that I am aware of would suggest that *stadtmanae* held the monopoly.

Despite the fact that *M. stadtmanae* holds the monopoly, Chibani *et al.* (2022) showed that other yet uncultured methanogens including *Methanosphaera* species remain to be discovered and characterized. Please see lines 79 to 96.

6. Line 73: It is not clear in the methods what primers were used to initially screen stool samples.

We added to appropriate reference from Drancourt *et al.*, 2020) in materials and methods section : “A 0.5-g 16S rRNA RT-PCR- positive (primers :Metha_16S_2_MBF, 5'-CGAACCGGATTAGATACCCG-3' and Metha_16S_2_MBR, 5'-CCCGCCAATTCCTTTAAGTT-3'; probe: FAM_Metha_16S_2_MBP, [FAM]-5'-CCTGGGAAGTACGGTCGCAAG-3') (40) stool sample was suspended in 10 mL of Dulbecco's Phosphate Buffered Saline (DPBS) 1X water (ThermoFischer Scientific, Waltham, Massachusetts, United States) and vortexed for a few seconds.” **(Lines 272-278)**.

7. Line 78: Methanosphaera spp. are Gram-positive, are authors sure that the Gram-variable cells were not a contaminant?

Gram-variable feature was described by Ferrari *et al.* (1994) for *M. oralis* older culture. *Ca. M. massiliense* is Gram-positive and becomes Gram-variable. The purity of the culture was ensure by autofluorescence and electron microscopy performed on several fields from the same culture used for Gram-colorations and furthermore we checked that no bacteria grow on COS agar plate.

The text was modified as follow:

“Colonies comprised of Gram-positive diplococci and clumps, with the individual diameter measured at 850-nm using electron microscopy. Gram-variability was observed after 15 days of culture. These non-motile cocci showed auto-fluorescence at 420 nm using confocal microscopy. The purity of the final culture was established as only autofluorescent cocci were observed in confocal microscopy from several fields as well as only cocci were observed in electron microscopy. Furthermore, we checked that no bacteria grow on COS agar plate.” **(Lines 105-112)**

8. Line 82: Following comment above are authors sure that the MALDI-TOF returned a nil result due to a database fail rather than the culture not being pure?

Proteic extraction was made from a pure culture. Spectra were firstly compared to a bacteria database then compared with *M. stadtmanae* DSM3091 (CSUR P9634) spectra and no significant difference was obtained.

The paragraph was modified as follow :

“MALDI-TOF mass spectrometry yielded reproducible peptide spectra but no identification by spectra comparison with our bacterial database. PCR-sequencing confirmed the identity of *Ca. M. massiliense*. Comparison with *M. stadtmanae* DSM3091 peptidic profile didn't show significant differences within the two spectra.” **(Lines 149-153)**

9. Line 85: (inc methods section) why did authors not use antibiotic strip testing for this assay? Authors report a lawn forming on agar so this should be possible and should yield a more accurate MIC?

The macrodilution technique was the reference technique used to test methanogen susceptibility in our laboratory (Dridi *et al.*, 2011) and culture on solid medium remained difficult and not always successful.

10. Line 86: rationale for these antibiotics?

We didn't expect to be surprised by the results and we wanted to confirm the susceptibility spectra of *Ca. M. massiliense*. The antibiotics selected and the concentration tested were based on known *M. stadtmanae* data (Dridi *et al.*, 2011).

11. Line 85: provide data/results/plots from "sensitive" method testing (AUC from GC-MS data) as supplementary

The data were added as supplementary in Supplementary Data 6.

12. Line 115: can authors please also include CheckM stats for BB6 (genome completeness and contamination percentages).

CheckM stats were made and gave 97.6 % completeness and 0% contamination. This was added in materials and methods and results section and in table 1. **Please see lines 122 and 334.**

13. Line 133: can authors describe origin and choice of massiliense as the species name in text here.

We provided information about the origin and choice of massiliense as the species name as follow :

« Following these phylogeny data and culture isolation, we introduced a new methanogen species named *Candidatus Methanosphaera massiliense* (*Ca. M. massiliense*) sp. nov. (Strain: BB6; CSUR number: Q7470), 'massiliense' is for 'Marseille', the city where it was isolated." **(Lines 145-148)**

14. Line 134: instead of referring to the sample as leftover stool please be more precise and describe the host e.g., "healthy xx-year-old male".

Unfortunately, as we used an anonymized leftover stool sample, we can't provide this information legally in a publication.

15. Line 135: if the strain has been publicly deposited, please describe this in text. If the strain is only held by the authors, then this is the 3rd *Methanosphaera* spp. to be isolated from humans (DSMZ3091, PA5 and BB6).

We modified the sentence as follow:

« This is the second representative *Methanosphaera* in humans deposited into the international public collection CSUR WDCM 875 (Collection Souches Unité des Rickettsies, WDCM 875, Marseille, France) under the reference Q7470 » **(Lines 200-202).**

16. Line 137: did authors generate the peptide fingerprint for *stadtmanae*? It would be worthwhile having this represented beside BB6 as a panelled figure for comparison.

We compared *Ca. M. massiliense* spectra with the peptide fingerprint of *M. stadtmanae* DSM3091 (CSUR P9634) and modify the text as follow, the figure (now Supplementary Data 5) was also modified:

“MALDI-TOF mass spectrometry yielded reproducible peptide spectra but no identification by spectra comparison with our bacterial database. PCR-sequencing confirmed the identity of *Ca. M. massiliense*. Comparison with *M. stadtmanae* DSM3091 peptidic profile didn't show significant differences within the two spectra.” **(Lines 149-153)**

17. Line 139: For me this not a very thorough description of the new species and is a basic in the sense that authors have not completed any substrate affinity assays to confirm the growth/methane production capabilities of BB6. Neither have the investigated the genome to report the molecular potential of BB6 to use different methanogenic pathways.

A substrate affinity assay was conducted, we also looked for formate dehydrogenase and enzymes for acetoclastic methanogenesis to confirm our results but we believe that research on the precise metabolism of *Ca. M. massiliense* should be the subject of further work.

Please see **lines 165-177** and Supplementary Data 6 for results and **lines 412-423** for methods.

Furthermore, our team initiated a new project on understanding the metabolic pathway of BB6 and other methanogens species belonging to *Methanosphaera* genus.

18. Line 143-152: this whole section belongs in the results and in some respects addresses my point above. As authors have BB6 in culture it is a simple task to complete a growth rate under each methanogenic substrate combination to validate findings seen in the genome.

We modified the results and discussion sections as follow:

Results: “Genome annotation revealed the presence of a NADPH-butanol dehydrogenase and an aldehyde dehydrogenase closely related to *Methanosphaera* sp. SH11033 and *Methanosphaera* sp. WGK6 (Supplementary Data 7 to 10), which allow growth in a hydrogen-free atmosphere in a medium containing methanol and ethanol. The absence of formate dehydrogenase and the presence of incomplete enzymatic pathway for acetoclastic methanogenesis, only the acetyl-CoA synthetase is encoded, are consistent with the previous experimental results.” **(Lines 171-177)**

Discussion: “Among isolated species, *Ca. M. massiliense* which was found to be most closely related to *Methanosphaera* sp. WGK6, a methanogen isolated by culture in one Western grey kangaroo (Figures 5, 6 and 7). Both methanogens are able to grow on methanol and hydrogen but also in an hydrogen-free medium containing methanol and ethanol as they share NADPH-butanol dehydrogenase and aldehyde dehydrogenase (28), GG medium supplemented with ethanol should be suitable for these methanogens growth.” **(Lines 213-219)**

A substrate affinity assay was also conducted, please see **lines 165-177** and Supplementary Data 6 for results and **lines 413-423** for methods.

19. Line 161: what was the occupation of the donor? Was it likely they would have had livestock exposure? What do authors mean by pig-derived foodstuffs? Are they referring to meat? I highly doubt methanogenic archaea (an oxygen sensitive microbe) would survive this amount of handling before ingestion.

Unfortunately, as we used an anonymized leftover stool sample, we can't provide these information legally in a publication as specified above in response 14. Pig-derived foodstuffs

are mainly referring to meat, highly consumed in South of France. Porks of this study were born in South of France and slaughtered in Corsica to be transformed in cold meat. We hypothesized that sources for *Ca. M. massiliense* may be mammalian zoonotic as food and animals may be a source for host transmission of microorganisms to the intestinal microbiota of humans.

We modified the paragraph as follow and we added new references:

“Indeed, through the results obtained in this study, we hypothesized that sources for *Ca. M. massiliense* may be mammalian zoonotic as suggested by the measured prevalence of *Ca. M. massiliense* which is 18 times higher in red kangaroos than in rarely exposed humans and 34 times higher in pigs whose populations in France are routinely exposed either by direct or indirect contact via pig-derived foodstuffs. Food and animals may be a source for host transmission of microorganisms to the intestinal microbiota of humans (32). Bacterial contamination of meat may occur at different steps from slaughtering to meat production and species involved depend on different parameters such as technical processes, hygienic conditions, storage temperature or gas packaging atmosphere (33–35). The contaminant may be part of the animal microbiota (36) but studies often focused on pathogenic or spoilage bacteria and methanogens contamination has never been investigated.” **(Lines 230-242)**

20. Line 177: humans are mammals, please rephrase this.

We modified the sentence accordingly:

« It is genetically more closely related to **mammalian animals** *Methanosphaera* than to *M. stadtmanae* » **(Lines 254 and 255)**

21. Line 184: please provide a description of the donor the isolate was generated from, at a minimum age, sex, health status (and ethnicity).

Unfortunately, as we used an anonymised leftover stool sample, we can't provide this information legally in a publication.

22. Line 200: as the previous publication has not been released please describe here how the SAB media was modified to create GG.

GG medium is now published, and we added to correct reference in the article. Please see **line 279**.

23. Line 206: what substrate sources were included? And what was the concentration ?

We used GG medium supplemented with ethanol (2% v/v), the substrates were acetate, formate, methanol, ethanol, fatty volatile acids and rumen.

We precised the text as follow:

“Then, 200 µL of growing culture was inoculated into a fresh **GG** medium containing fresh antibiotics, being 100 mg/L amoxicillin, 100 mg/L vancomycin, 100 mg/L imipenem, 50 mg/L daptomycin and 50 mg/L amphotericin B (Sigma Aldrich, Saint Quentin Fallavier, France) and **supplemented with ethanol at 2%** (v/v).” **(Lines 284-288)**

24. Line 294: what Illumina platform was used?

We used MiSeq Illumina platform, we modified the sentence as follow :

« DNA was then engaged for whole genome sequencing (WGS) using the **MiSeq** Illumina pair-end protocol (Illumina, San Diego, California, USA)” **(Line 327)**

25. Line 321-329: please describe qPCR cycle conditions here.

We added the PCR amplification program as follow:

“. The RT-PCR amplification program was 95°C for 15 min, followed by 39 cycles of 95°C for 30 s and 60°C for 1 min and a final step of 40°C for 30 s.” **(Lines 438-440)**

26. Line 340: this data deposit is only for the partial 16S gene. Please include the other accession IDs or if not completed yet please also deposit the raw sequence reads and WGS to NCBI and provide these accession IDs.

Accession IDs were added as follow:

« All genomic data were submitted to the GenBank NCBI database under accession number OP930955 (16S rRNA sequence) and JARBXM000000000 (genome). The amplicons sequences were also deposited under accession numbers: OX456088; OX456151; OX456152; OX456153; OX456154; OX456155; OX456156; OX456157; OX456158; OX456157; OX456158; OX456159; OX456160; OX456161; OX456162; OX456163; OX456164; OX456165; OX456166; OX456167; OX456168; OX456169; OX456170. ». **(Lines 461 and 467)**

27. Figures: resolution of most images are very poor and unsuitable for publication, figure legends also need to be clearer and figures which are panels need to be formatted accordingly before upload.

All the figures and legends were improved, we hope that they will be suitable for publication.

28. Figure 1: significantly reduce white space in this image. I would even recommend moving this to supplementary material as it contains only generic images and no results generated by the authors.

The figure was totally remastered, and additional information were given in legend. The figure was also moved in supplementary as Supplementary Data 1.

29. Figure 2: this image needs to be paneled and labelled with A-C before reupload. Resolution needs to be improved. Scale bar needs to be included on 2A and cropped to included only those cells in focus. The scale of the image also makes it near impossible to see the cells without having to zoom in. 2B can also be cropped to remove dead space. 2C comes up as 3 images on my manuscript copy and is poor quality. Is this image to show the autofluorescence? If so, a better quality image needs to be taken before resubmission. Figure legend can also be more descriptive.

The figure (now figure 1) was totally remastered as one panel (A-F) with new images, scale bars were added for all images and additional information was given in legend.

30. Figure 3: legend needs to be more descriptive so anyone not familiar with MALDI-TOF output can interpret the results. I would also recommend paneling this with a comparison of the DSMZ3091 profile.

The figure 3 (now Supplementary Data 5) was totally remastered and *M. stadtmannae* spectra was added. The figure was moved to supplementary as Supplementary Data 5 as asked by the other reviewer.

31. Figure 5: line 477-483: this a mixture of methods and results. Please condense this information in the figure legend and provide in the results and methods section where appropriate.

The figure 5 (now figure 3) was modified and now includes the 100 hit blasts. The legend was also modified, and the information was added in the appropriate sections. Please see **lines 341-351** for methods and **lines 130-134** for results.

Reviewer #3

General remarks

- The article does not represent a valid description of the new culture. Therefore, the isolate should be referred to as "Candidatus *M. massiliense*" throughout.
- In the spirit of Open Science, please provide the accession number for the genome in question, not just for the 16S rRNA gene.
- The new isolate should be included in a public culture collection and made available to all researchers.
- The genome sequence was not compared to the largest source of data for archaeal genomes from the human gut (Chibani et al., 2022); this needs to be included to put the result in context.
- The zoonotic prevalence of *Cand. M. massiliense* is a very strong statement and requires additional confirmation through appropriate assessment of the approach (i.e., sequencing of RT-PCR products or metagenomics) and inclusion of ct values for all samples and controls and calculation of copy numbers.

The authors would like to thank the reviewer for his comments. We hope that this remastered manuscript will give entire satisfaction to the reviewer especially genomic and RT-PCR analysis. The genome sequence is now available on NCBI database as well as the strain has been included in the public collection WDCM 875.

Specific remarks

- **Line 13:** running head is not appropriate, zoonotic origin has not been shown.

We change the running title by: « *Candidatus Methanosphaera massiliense* sp. nov. » **(Line 13)**

- Please check the title, it does not make sense and is probably incorrect (questions ?)

We change the title by : « *Candidatus Methanosphaera massiliense* sp. nov. questions transfer between human and animal microbiota. » **(Lines 1 and 2)**

- Line 24: This is not correct. Next to *M. stadtmannae*, also other *Methanosphaera* representatives have been detected via molecular methods, e.g. by genome binning. Please check Chibani et al. 2022.

The whole paragraph was modified as follow, including corrections asked by the reviewer 2:

“Whereas *M. stadtmannae* was for a long time the sole *Methanosphaera* member cultured from human faeces (5) and further detected by PCR-based methods in human dental plaque (25), a new human isolate, *Methanosphaera* sp. PA5 was provided by Hoedt *et al.* in 2018 (26). Furthermore, *Methanosphaera* have been cultured from mammal guts including *Methanosphaera cuniculi* (*M. cuniculi*) in rabbit (27), *Methanosphaera* sp. WGK6 in Western grey kangaroo (*Macropus fuliginosus*) (28) and *Methanosphaera* sp. BMS in cow guts (26). Recently, Hoedt *et al.* (26) and Chibani *et al.* (9) obtained respectively 7 and 20 *Methanosphaera* spp. genomes from metagenomic data of mammals (including humans) gut as reported in **supplementary data 4**, some of them not corresponding to known isolates. Particularly, among the 20 *Methanosphaera* spp. genomes provided by Chibani *et al.* from the human gut, 17 were related to *M. stadtmannae*, two to *M. cuniculi* and one to a genome

obtained from cow rumen, suggesting possible host transfers and adaptation from animals to human gut. (Lines 79-92)

- Line 27 here and everywhere: should be *Candidatus M. massiliense*

« *Candidatus* » or « *Ca.* » was added everywhere needed within the text.

- Line 30: rephrase, a gap is not expected in a microbial genome? You mean you were able to produce a gapless genomic sequence, correct?

The authors thank the reviewer for the highlighted point and the authors clarify the point:

According to D-fast analysis, there is no gap. D-fast link:

https://dfast.ddbj.nig.ac.jp/help_annotation.

The authors corrected the sentence below to clarify the review comment:

“We were able to produce a 29.7% GC content, gapless 1,785,773-bp genome sequence with an 84.5% coding ratio; encoding for alcohol and aldehyde dehydrogenases promoting *Ca. M. massiliense* growth without hydrogen.” (Lines 30-33)

- Line 33 novel screening systems require proper evaluation, e.g. sequencing of the qPCR results, otherwise these statements cannot be made.

All the amplicons were sequenced and « *Ca. M. massiliense* identity in all positive samples was confirmed as the sequences showed an identity average of 96.28% +/- 0.82 and 100% coverage with *Ca. M. massiliense* whole genome sequence and as no other correspondence was found in GenBank database » (Lines 191-195). Please see lines 451-459 for methods. The amplicons sequences are in Supplementary Data 14 and were deposited on GenBank database.

- Line 36: Why does your finding question the possible zoonotic source? It rather opens the questions on the origin... Maybe rephrase, the meaning is not clear.

We rephrased as follow:

“This study, extending the diversity of *Methanosphaera* in human microbiota, opens the question on the origin of *M. massiliense* and possible transfer between hosts.” (Lines 35 to 37)

- Line 39: the only cultivated *Methanosphaera* representative

We added the word « cultivated » in the sentence as follow:

« Methanogens are constant inhabitants in the human gut microbiota in which *Methanosphaera stadtmanae* was the only cultivated *Methanosphaera* representative. » (Lines 39 and 40)

- Line 42: This is a strong claim and has to be supported by e.g. proper validation of the qPCR primers by sequencing, or metagenomic sequencing of the samples used.

All the amplicons were sequenced and « *Ca. M. massiliense* identity in all positive samples was confirmed as the sequences showed an identity average of 96.28% +/- 0.82 and 100%

coverage with *Ca. M. massiliense* whole genome sequence and as no other correspondence was found in GenBank database » **(Lines 191-194)**. Please see **lines 451-459** for methods. The amplicons sequences are in Supplementary Data 14 and were deposited on GenBank database.

- Line 50: also other members have been isolated, namely *Cand. M. intestini*, *Cand. M. intestinalis*, *Cand. M. alvus*. Please update your references.

We added the three methanogens with the appropriate references as follow :

« *Candidatus Methanomethylophilus alvus* (6), *Candidatus Methanomassiliicoccus intestinalis* (7) and *Candidatus Methanobrevibacter intestini* (8–10) » **(Lines 53 to 55)**

- Line 69: it is not clear whether it is novel per se, as its sequence was not put in context with retrieved genomic bins from human gut (again, see Chibani et al).

Ca. M. massiliense clustered with *Methanosphaera* sp. RUG761 (ANI, 99%) and *Methanosphaera* sp. M5 (ANI, 99%), that clustered together (ANI, 99%), and with *Methanosphaera* sp. SHI1033 (ANI, 97%). The dDDH values with these three species were above 70% whatever the DDH calculation method. All these data suggested that *Ca. M. massiliense* might be their cultured representative.

Please see results **lines 130-144** and figure 4, discussion paragraph **lines 208-212**.

All the genome sequences used in this study are reported in Supplementary Data 4.

We also added the word "isolate" **line 99**.

- Line 74/133: consider removing "leftover" here, as this is confusing for the reader. It is explained in the materials and methods.

We removed the word « leftover » in the two sentences as follow :

« Methane was detected from one methanogen 16S rRNA gene PCR-positive stool sample culture. » **(Line 102)**

« This is the second representative *Methanosphaera* in humans deposited into the international public collection CSUR WDCM 875 (Collection Souches Unité des Rickettsies, WDCM 875, Marseille, France) under the reference Q7470 » **(Line 200)**.

- Line 80 : showed auto-fluorescence

We modified the sentence as suggested:

« These non-motile cocci **showed auto-fluorescence** at 420 nm using confocal microscopy » **(Line 108)**.

- Line 81: Was the purity of the final culture checked, and how?

The purity of the culture was ensure by autofluorescence and electron microscopy performed on several fields from the same culture used for Gram-colorations and furthermore we checked that no bacteria grow on COS agar plate.

The text was modified as follow:

“The purity of the final culture was established as only autofluorescent cocci were observed in confocal microscopy from several fields as well as only cocci were observed in electron microscopy. Furthermore, we checked that no bacteria grow on COS agar plate.” **(Lines 109-112)**

- Line 97: 20 different Methanosphaera genomes representing unique strains were detected in Chibani et al., please put your genome in context (ANI differences, tree)

We added new data including those from Chibani *et al.*, all the genome sequences used in this study are reported in Supplementary Data 4. Please see results **lines 130-144** and figure 4, discussion paragraph **lines 208-212**.

- Please give further information on the genomic sequence, e.g. the completeness of marker genes, and contamination **(e.g. use CheckM)**

CheckM stats were made and gave 97.6 % completeness and 0% contamination. This was added in materials and methods and results section and also in table 1. **Please see lines 122 and 334.**

- Line 106: cover = coverage?

The word « cover » was replaced by the word « coverage » as follow :

« The 16S rRNA gene sequence from the whole genome shared 97.22% sequence similarity (100% **coverage**) with *M. stadtmannae*, 96.95% (100% **coverage**) with *Methanosphaera* sp. BMS, 96.55% (99% **coverage**) with *M. cuniculi* and 98.9% (92% **coverage**) with *Methanosphaera* sp. WGK6. » **(Lines 126 to 130)**

- Line 114: Candidatus

The word Candidatus was added, please see **line 146**.

- Line 114: please provide genome sequence in a public repository; also the culture should be placed in a public collection

The genome sequence is now available on NCBI database under accession number JARBXM0000000000 and the strain has been included in the public collection WDCM 875 under the reference Q7470.

- Line 118: evaluation of the new qPCR primers and set-up has to be done more thoroughly.

All the amplicons were sequenced to confirm the specificity of the PCR system and « *Ca. M. massiliense* identity in all positive samples was confirmed as the sequences showed an identity average of 96.28% +/- 0.82 and 100% coverage with *Ca. M. massiliense* whole genome sequence and as no other correspondence was found in GenBank database » **(Lines 191-195)**. Please see **lines 451-459** for methods. The amplicons sequences are in Supplementary Data 14 and were deposited on GenBank database.

- Line 122: please report quantitative data, not only qualitative information (presence vs. non-present). It is important to know the relative abundance, not only the prevalence.

Quantitative data were added. Please see **lines 448-450** for methods and **lines 195-197** and Supplementary Data 13 for results.

- Line 129: what is the purpose of this Nanoarchaea research chapter?

We removed this chapter.

- Line 135: please give details: to which public collection was it submitted? And, it is not thoroughly described: no proper description is made. <https://www.microbiologyresearch.org/content/journal/ijsem?page=about-journal#2>

The strain has been included in the public collection WDCM 875.

We modified the sentence as follow :

« This is the second representative *Methanosphaera* in humans deposited into the international public collection CSUR WDCM 875 (Collection Souches Unité des Rickettsies, WDCM 875, Marseille, France) under the reference Q7470 » (**Lines 200-202**).

We also added new features about *Ca. M. massiliense* including phenotypical description, pH and antibiotics susceptibility, peptide fingerprint spectra, substrates affinity assay and whole genome sequence-derived analyses.

- Line 136: do you mean light and electron microscopy?

The word « optic » was replaced by « light » everywhere needed within the text.

- Line 138: this does not mean it is properly evaluated

We hope the new data will give entire satisfaction to the reviewer.

- Line 146ff: Please provide the genome sequence in a public database. Until this is not done, such statements are not useful to any reader

Accession IDs were added as follow :

« All genomic data were submitted to the GenBank NCBI database under accession number OP930955 (16S rRNA sequence) and JARBXM000000000 (genome). » (**Lines 461-462**)

- Line 157: did you really check whether it remained alive? Or was this the range of cultivation?

We subcultured *Ca. M. massiliense* from the two pH conditions and it yielded to methane production after a 7-days culture. Please see **lines 161-164** and Supplementary Data 6 for results and **lines 411-412** for methods.

- Line 200: The composition of the medium is crucial to this publication. Either, include a citation or describe the medium.

GG medium is now published, and we added to correct reference in the article. Please see **line 279**.

We used GG medium supplemented with ethanol (2% v/v) for the final culture, the substrates were acetate, formate, methanol, ethanol, fatty volatile acids and rumen.

- Line 210: how exactly was the purity ensured?

The following sentences were added in the results and methods sections to answer the review's question:

"The purity of the final culture was established as only autofluorescent cocci were observed in confocal microscopy from several fields as well as only cocci were observed in electron microscopy. Furthermore, we checked that no bacteria grow on COS agar plate." **(Lines 109-112)**

"The isolate was subcultured ten times to ensure the purity of the culture, checked by autofluorescence using a confocal light microscope LSM 900 (Carl Zeiss Microscopy GmbH, Jena, Germany) at the 63 X objective with immersion oil as previously described (43), electron microscopy with the Scanning Electron Microscope TM4000 plus (Hitachi, Tokyo, Japan) and inoculation of 100 μ L on COS agar plate." **(Lines 294-299)**

- Line 341: the accession number does not refer to the submission of the genome. This only includes the 16S rRNA gene.

Accession IDs were added as follow:

« All genomic data were submitted to the GenBank NCBI database under accession number OP930955 (16S rRNA sequence) and JARBXM000000000 (genome). » **(Lines 461 and 462)**

- Table 1: Is the genome closed or not? Indicate contamination and completeness of the genome

CheckM contamination and completeness were added in the table 1.

- Fig.1: explain what is shown on the figure. What is the picture on the right?

The figure was totally remastered, and additional information were given in legend. The figure was also moved in supplementary as Supplementary Data 1 as asked by the other reviewer.

- Fig.2: arrange all pictures for Fig. 2 in one panel
- Fig. 2a should be cropped and has to include a bar.
- Same applies to Fig. 2b, provide more details on the technique itself (negative stain, shadowed... etc.).

The figure (now figure 1) was totally remastered as one panel (A-F) with new images, scale bars were added for all images and additional information were given in legend.

The negative stain was due to PTA (PhosphoTungstic Acide) addition, living cells have a light stain and dead ones have a negative stain. PTA was not added to the new preparation, so all the cells have the same color.

- Move Fig. 3 to supplementary

The figure (now Supplementary Data 5) was totally remastered and *M. stadtmannae* spectra was added as asked by the other reviewer. The figure was moved to supplementary as Supplementary Data 5.

- Fig. 4: add more explanation to the figure itself (legend?)

The legend of the figure 4 (now figure 2) was modified as follow to provide more explanation:

« **Figure 2: Circular representation of *Candidatus Methanosphaera massiliense* sp. nov. genome.** Genome representation was performed using the Proksee platform with standard parameters (version 1.0.0), (<https://proksee.ca/>). The two contigs are presented with color-coded highlights to distinguish important genomic features and provide information about the composition and structure of *Candidatus Methanosphaera massiliense* sp. nov. genome. ORF regions are indicated in blue, GC content is in black, and positive and negative GC skew in green and red respectively. This analysis reveals the distribution of protein-coding genes and the variation in GC content across the genome. » **(Lines 668-676)**

- Fig. 7: what is an atomic representation?

Atomic representation was a representation of dDDH value with *Ca. M. massiliense* and its prevalence in animals as an atome with electrons arranged in layers. This figure was removed as too much data should have been included according to the new results and would have risked complicating understanding.

Sincerely,
Pr Elodie TERRER

May 22, 2023

Prof. Elodie Terror
Aix-Marseille Universite
MARSEILLE
France

Re: Spectrum05141-22R1 (*Candidatus* Methanosphaera massiliense sp. nov. questions transfer between human and animal microbiota.)

Dear Prof. Elodie Terror:

Link Not Available

Sincerely,

Henning Seedorf

Journals Department
Reviewer comments:

Reviewer #2 (Comments for the Author):

I thank the authors for addressing my initial comments, below are a few additional comments to clarify the added material.

1. Line 151: remove "PCR-sequencing confirmed Ca. M. massiliense." or reformat to specify that the colony/cell mass was first confirmed through PCR-sequencing to match BB6 isolate before MALDI-TOF analysis. Otherwise this reads a little disjointed.
2. Line 161: replace "the methanogen" with isolate BB6.
3. Line 165: can authors please also include a graph of the growth rate (optical density) over time. This would be valuable to readers particularly anyone who wishes to request your new isolate, as it will give them an example of what to expect of the culture time and yield.

4. Line 167: this is misleading, authors need to specify that it was ethanol+rumen fluid+VFA resulted in methane production (without going to the supp data readers wouldn't know that ethanol in GG alone results in no methane).
5. Line 168: please include growth curves for each substrate tested as a supplementary figure.
6. Line 167: I'm a little confused by the substrate analysis. GG+ethanol= no methane but GG+VFA+rumen fluid+ethanol = methane. I would suggest if authors want to include this data they need to make some effort into working out what carbon source is being carried over from the rumen fluid or what VFA is being used as the carbon donor for methanogenesis? Did authors complete any analysis with just rumen fluid and/or VFA added alone (please report these controls).
7. Supp data 6 - please provide scale of measurement for these values in the tables and in the table legend. If possible, I would suggest converting (what I assume is AUC) to molar concentrations for each gas based on your standard curve.
8. Line 201: rephrase "this is the second representative Methanospaera species". Transfer the deposition information to the results.
9. Line 210-213: if BB6 might be a cultured representative of RUG761 and M5 how is it most closely related to WGK6? Please rephrase, perhaps authors mean to say BB6 is metabolically most closely related to WGK6 the only other Methanospaera demonstrated in culture to utilise methanol+H₂ and methanol+ethanol.
10. Line 272-275: the inserted text is in the wrong location.
11. Line 279: GG medium: it is still unclear from the reference and this manuscript about the recipe. It appears acetate, formate and VFAs are added as standard (according to ref 41)? And no mention of rumen fluid.
12. Line 287: if the GG medium was only supplemented with ethanol this does not align with the substrate analysis results. Did the isolation medium also contain rumen fluid and VFA? There is no reference of rumen fluid in either ref 41 or the SAB ref. In fact I wonder if acetate/format were present and ethanol was added as an electron donor at time of isolation was it using this as the carbon donor? The substrate analysis does not answer this question. Can BB6 use something other than methanol with ethanol? The results would suggest this is a strong possibility, it is an exciting observation, but it has not been addressed at all by the authors.
13. Line 299: what is the COS agar recipe this has not be described?
14. Line 414: authors mention rumen fluid, but it has not been described anywhere else in the methods. What breed was it collected from, what was it eating and how was this prepared for the medium, what % was added?
15. Line 419: remove "with methanol (2% v/v) or ethanol (2% v/v)" this looks like a repeat from lines 416-417?
16. How did authors determine the CFU/mL? is this based on optical density? Please report.
17. Figure 1: gram stain image is not super clear; resolution could be improved.
18. Figure supp data 1: please include in this figure the addition of substrates for isolation (i.e. ethanol) without reading the legend the figure would imply you isolated BB6 on nitrogen alone. The picture of the agar plate still is not very clear, were authors able to pick a single colony? Was this spread or streaked on agar during the isolation process? A close up pic of a colony might be a better inclusion for this image so readers can see what it looks like morphologically?

Reviewer #5 (Comments for the Author):

The main point of the manuscript from Pilliol et al is the isolation (but relatively poor description) of a new species from the genus Methanospaera, obtained from a human stool sample. The importance and diversity of archaea in humans, and in human gut microbiota (mainly methanogens) has been highlighted by several and sometimes important reviews on this topic. Having cultured representatives besides (meta)genomic data is an important step for addressing physiological or ecological questions. However, I feel frustrated after reading and analysing this manuscript (the revised version is the only version that I have reviewed), concerning several data given by authors. This is described in more details below. Of all, some elements are not clear by lack of details and the title sounds mysterious to me as it is said that this isolate "questions transfer between human and animal microbiota" : This seems of course new about this isolate, but not really new compared to some other and sometimes old papers (eg. archaeal DNA / sequences, even organisms seen from human studies like Haloarchaea, other orders of methanogenic archaea and other species from Methanobacteriales which are found in ruminants, pigs, ...). This could be reasonably forgotten if the manuscripts provided more specific data to give some other clues supporting a transfer from animal / foodstuff rather than a unique human subject and prevalence in some animals. Some of these points are detailed below. It is of course very likely that this isolate belongs to a host-associated group and not an environmental one (like other Methanospaera spp and as it can be inferred from Supp Data 3 in which no «real» environmental sequences are retrieved from the first 100 hit blasts - this is an important point for readers that could be clearly disclosed in the manuscript). Also, I strongly recommend authors to send their isolate to a second collection of organisms (ATCC, DSMZ, JCN,...) to ensure it to be safe among time.

Specific comments and questions arising from the reviewing :

1. The isolation relies on a new medium called "GG medium" : it has been recently published by the author (ref 41 in the manuscript) and indicated as patented ("conflict of interest section"). As its publication is recent, it would be interesting to give its composition in the manuscript, either in M&M section or added to a figure legend (eg legend of Supp Data 1). Moreover, as I could not retrieve the recipe from ref 41, it is very unclear of what is present and at which quantity within this growth medium: eg, does it contain volatile fatty acids and filtered rumen fluid as suggested by supp data 6 "substrate affinity test" ? If yes, which ones ? It could be important for the reader to know, considering the results observed for substrate affinity tests.
 - o About the isolation and description of this new species, it would be great to conform to some standards. The growth optimal pH

is defined at 7.3, only because there is a growth at pH6 and pH7.3 and not at pH5.0 and pH8.0, which does not seem sufficient to claim 7.3 as optimal pH.

o What is the sensitivity to O₂, and micro-aerophily of this isolate, its viability in aerobia environment ? Could you test it in order to give any support to your hypothesis of transfer from foodstuff ?

2. Parts about the human subject who is positive for the isolate are very unclear for me and has to be clarified :

o It is written lines 189-191 that "the one positive human faeces sample had been collected from the same individual from whom *Ca. M. massiliense* had been isolated by culture as above". This means that a 2nd collect of faeces has been done ? Could you indicate, even approximately, the time passed between the two sampling dates (days, weeks or months ?) in order to sustain a gut-inhabiting microbe in this subject? Or is it a miswording as it is written in M&M "specimen collection" part, that anonymized leftovers were used.

o In fact, it has to be clearly established for the reader if this species has been retrieved twice independantly from the same donor or not. If yes, isn't it possible (ethically) to give at least her/his age/gender/ethnicity ? Ideally also, other informations like health status and profession.

3. Detection of the isolate has been performed in stools of several animals. From data given in Supp Data 13 :

o while the qPCR detection test seems ok as it is described and how itw as implemented, how can be explained such a discrepancy between Ct values and CFU/mL observed in the human subject compared to all other positive animals ? Eg, the CFU values in animals would theoretically lead to Ct values about 24-25 (and not 32-33 as observed), if compared to the values observed in the unique human. In fact, I am wondering about the real quantity / proportion of this isolate in the human gut microbiota. Has the qPCR assay a good linearity over the quantities which are tested ?

o these data clearly indicate that this species is far more prevalent in red Kangaroos and pigs (and not present in other animals that were tested) but far less in quantity in all these animals than in the unique human. Could you indicate approximately to which proportion it corresponds among other faecal organisms, in this human subject, compared to other animals ? In the human subject, is it associated with other methanogens ?(at which comparative abundance and which ones ?)

4. besides the isolation of this *Methanospheara* sp. strain, the authors highlight that this occurrence in a human subject arises likely from a transfer from animal (pigs, less likely kangaroos, as it was not detected in any other animal tested, ie dogs, cows, sheep and horses), and that could occur from foodstuff. Even if such a zoonotic transfer hypothesis is likely, I would omit the text from lines 230 to 242, just keeping "Indeed, through.... As suggested by prevalence of *Ca. M. massiliense* in some animals." (I would however keep the last paragraph of the conclusion as it stands). Lines 232 to 240 seem to me out of the discussion.

Other minor points :

- abstract :

o please change line 35 "...and 0.67% in human faeces" by "and no detection in 149 (or 150 ?) other human samples"

o the word «origin» line 36 is ambiguous. Please rephrase.

- depending on answers to point 2 and 3, please remove "and adaptation to human gut" (line 45) as evidences seem lacking to claim that this species is inhabiting the gut of this human subject.

- Line 429 and after: please indicate in the text if in silico, the qPCR test should be positive for strain RUG761 (and also strain Mb5) ?

- Line 223 "implanted" instead of "implanting" ?

- Lines 230-235: please rephrase and omits numerical values as it does not make sense with only one positive human sample.

- Line 685, "calculated FROM THE using PYANI...."

Thank you for this manuscript. The isolate may help in disclose more precisely some aspects of methanogenesis, especially linked to alcohols.

Staff Comments:

Preparing Revision Guidelines

For complete guidelines on revision requirements, please see the journal Submission and Review Process requirements at <https://journals.asm.org/journal/Spectrum/submission-review-process>. **Submissions of a paper that does not conform to**

Microbiology Spectrum guidelines will delay acceptance of your manuscript. "

Please return the manuscript within 60 days; if you cannot complete the modification within this time period, please contact me. If you do not wish to modify the manuscript and prefer to submit it to another journal, please notify me of your decision immediately so that the manuscript may be formally withdrawn from consideration by Microbiology Spectrum.

Dear authors,

The objective of this study was to partially characterize the *Methanosphaera* species derived from human feces. The authors proposed the potential transmission of this *Methanosphaera* species from animals to the human gastrointestinal tract, although the evidence supporting this claim was not compelling.

Major concerns and general comments

- 1) In this study, a *Methanosphaera* species was isolated using the GG medium, which was originally designed with formate and acetate as substrates and specifically intended for *Methanobrevibacter smithii*. According to the substrate affinity test, this *Methanosphaera* isolate grows on methanol under H₂/CO₂/N₂ atmosphere. However, it is unclear why methanol was not used in the initial isolation procedure. Can you provide clarification on this matter? You have only mentioned a reference for the GG medium. Briefly mention the major ingredients of this GG medium. It will be helpful for the readers.
- 2) Was the confocal light microscope the sole method used to ascertain the purity of the culture? How was the potential contamination with other methanogen species? While autofluorescence and cocci morphology were utilized to confirm the purity of the isolate, it should be noted that all *Methanosphaera* species share these characteristics. How did you ensure that the culture exclusively contained the specific isolate?
- 3) Based on the phylogeny derived from the whole genome sequences, the current isolate exhibited close clustering with rumen *Methanosphaera* species (RUG761, SHI1033). However, the prevalence of these species in rumen samples was not examined in this study. Additionally, it would be intriguing to investigate whether the inclusion of rumen fluid in the GG medium enhanced the growth of this isolate. This investigation would provide valuable insights into the adaptability of the isolate to its specific environment.
4. The media used for the pH susceptibility, antibiotic susceptibility, and substrate affinity test lacked clarity. Please provide the information on the media and substrates used. It was mentioned in the text (lines 413-415), that basal media without acetate, formate, volatile fatty acids, and rumen fluid were used for the substrate affinity test. However, supplementary data 6 mentioned that GG medium was supplemented with Ethanol (containing volatile fatty acids and rumen fluid). Therefore, please provide clarification on the specific media and substrates used for each test.
5. What was the rationale behind selecting a 7-day time frame for assessing methane formation? Additionally, it would be beneficial to include growth curves for each

substrate. This would provide a more comprehensive understanding of the temporal dynamics and trends associated with growth/methane production.

6. pH susceptibility test: The authors conducted the test using a wide range of pH values (2-10), but specific pH requirements were not clearly defined in the current study.
7. Why didn't you test the optimal temperature range for the growth of this isolate?
8. Another significant limitation of this study is that much of the discussion surrounding the results appear to be more speculative rather than based on concrete evidence. The lack of appropriate references to support these speculations is also a notable drawback. See below:

Lines 222-225- Methanogens primarily inhabit the colon of human GIT. Methanol derived from pectin fermentation serves as the primary substrate for Methanosphaera. The claim made about presence of methanol from beverages and food reaching the colon requires clarification. Are you referring to the levels of methanol/ethanol in the bloodstream here?

Lines 230-242: It has been proposed in this study that the methanogens isolated could potentially transfer from other mammals based on their prevalence in the feces of mammals. However, relying solely on prevalence is insufficient to support this suggestion. Methanogens are strictly anaerobic organisms, and their ability to contaminate food materials requires clear logical explanation. Gut microbes are known to be highly specific, making it difficult for other microbes to colonize easily. Even if they manage to colonize, their ability to persist in the gut for an extended period of time is limited. The fact that the same isolate was obtained from the same individual at two different time points suggests that it has successfully colonized the gut. Further clarification is needed to support the claim of a potential transfer from other mammals.

Specific comments

- Line: 96: what is the recently designed hydrogen-free carbon-dioxide-free medium: Give a reference or the name of the media here.
- Line 315: culture or culture supernatant?
- Line 414: volatile fatty acids and rumen fluid
- Figure 1: this figure especially the electron microscopy figures B and C are not clear enough to observe specific features of the cell.
- Table 1: Can you expand the 2nd column length?

- Supplementary data 3: Mention the 16S rRNA gene sequence in the title.

Reviewer #2 (Comments for the Author):

I thank the authors for addressing my initial comments, below are a few additional comments to clarify the added material.

Authors 'answer: The authors would like to thank the reviewer for his additional comments and hope this new revised version will give entire satisfaction to the reviewer.

1. Line 151: remove "PCR-sequencing confirmed *Ca. M. massiliense*." or reformat to specify that the colony/cell mass was first confirmed through PCR-sequencing to match BB6 isolate before MALDI-TOF analysis. Otherwise, this reads a little disjointed.

Authors 'answer: The sentence was modified according to reviewer's recommendation:

"The cell mass was first confirmed through PCR-sequencing to match BB6 isolate before MALDI-TOF analysis." (Lines 155-156)

2. Line 161: replace "the methanogen" with isolate BB6.

Authors 'answer: The sentence was modified accordingly line 166.

3. Line 165: can authors please also include a graph of the growth rate (optical density) over time. This would be valuable to readers particularly anyone who wishes to request your new isolate, as it will give them an example of what to expect of the culture time and yield.

Authors 'answer: The authors provided a growth follow-up curve of methane production and optical density in Table S4.

4. Line 167: this is misleading, authors need to specify that it was ethanol+rumen fluid+VFA resulted in methane production (without going to the supp data readers wouldn't know that ethanol in GG alone results in no methane).

Authors 'answer: The authors apologize for the misunderstanding and clarified the paragraph as follow:

"*Ca. M. massiliense* was able to grow on GG0 medium (basal GG medium) with methanol under H₂/ CO₂/ N₂ atmosphere, on GG0 medium with methanol and ethanol under nitrogen atmosphere and on GG medium (containing acetate, formate and methanol) supplemented with ethanol as the reference condition (Table S4). Proxi-growth curve of methane production and optical density from day 0 to day 7 are also available in Table S4." (Lines 171-177)

The name « GG0 medium » was introduced in the materials section and referred to the basal GG medium used in this study:

"A basal GG medium (GG0) derived from the original GG medium but without acetate, formate nor methanol or ethanol was used, and each substrate or substrate combinations were added separately in the medium GG0. Nine conditions were tested using GG0 medium, six under nitrogen atmosphere: no substrate, acetate (2 g/L), formate (2 g/L), methanol (2% v/v), ethanol (2% v/v), methanol and ethanol (2% v/v) and three under 5% H₂/ 20% CO₂/ 75% N₂ atmosphere without any substrate, with methanol or with ethanol." (Lines 424-430)

5. Line 168: please include growth curves for each substrate tested as a supplementary figure.

The authors added histograms of methane production at day 7 for each substrate in **Table S4**.

6. Line 167: I'm a little confused by the substrate analysis. GG+ethanol= no methane but GG+VFA+rumen fluid+ethanol = methane. I would suggest if authors want to include this data they need to make some effort into working out what carbon source is being carried over from the rumen fluid or what VFA is being used as the carbon donor for methanogenesis? Did authors complete any analysis with just rumen fluid and/or VFA added alone (please report these controls).

Authors 'answer: The authors apologize for the mistake, GG medium as published in the patent FR 23 01404 doesn't contain volatile fatty acids (VFAs) nor rumen. The authors modified Table S4 data accordingly. Thereby, we didn't think it was necessary in this study to test VFAs and rumen as controls. However, it would be interesting to test in further studies and the authors thank the reviewer for his comment. GG0+ethanol under nitrogen atmosphere, GG0 under H₂/CO₂/N₂ atmosphere and GG medium supplemented with ethanol led to methane production.

Authors think that the important point was to find an optimal medium for the methanogen growth and thus, we didn't test other substrates combinations.

The authors clarified the paragraph as follow:

"*Ca. M. massiliense* was able to grow on GG0 medium (basal GG medium) with methanol under H₂/ CO₂/ N₂ atmosphere, on GG0 medium with methanol and ethanol under nitrogen atmosphere and on GG medium (containing acetate, formate and methanol) supplemented with ethanol as the reference condition (Table S4). Proxi-growth curve of methane production and optical density from day 0 to day 7 are also available in Table S4." (**Lines 171-177**)

7. Supp data 6 - please provide scale of measurement for these values in the tables and in the table legend. If possible, I would suggest converting (what I assume is AUC) to molar concentrations for each gas based on your standard curve.

Authors 'answer: The authors would like to clarify that the data presented in the article fully address our scientific question.

The authors have specified in all relevant sections that it refers to the area under the curve (AUC) and converted the data in molar concentration (M, mol/L) in **Table S4**.

8. Line 201: rephrase "this is the second representative *Methanosphaera* species". Transfer the deposition information to the results.

Authors 'answer: The deposition information were transferred in results section (**Lines 150-152**) and the sentence was rephrased as follow: "*Ca. Methanosphaera massiliense* is the second *Methanosphaera* species isolated in humans" (**Lines 210-211**).

9. Line 210-213: if BB6 might be a cultured representative of RUG761 and M5 how is it most

closely related to WGK6? Please rephrase, perhaps authors mean to say BB6 is metabolically most closely related to WGK6 the only other *Methanosphaera* demonstrated in culture to utilise methanol+H₂ and methanol+ethanol.

Authors 'answer: The authors had precised that is was among isolated species (**Line 221**) and added the word "metabolically" (**Line 222**).

10. Line 272-275: the inserted text is in the wrong location.

Authors 'answer: The text was moved in the "Sample collection" section as follow :
"Samples were screened for methanogens presence using RT-PCR (primers :Metha_16S_2_MBF, 5'-CGAACCGGATTAGATACCCG-3' and Metha_16S_2_MBR, 5'-CCCGCCAATTCCTTTAAGTT-3'; probe: FAM_Metha_16S_2_MBP, [FAM]-5'-CCTGGGAAGTACGGTCGCAAG-3') as previously described (40)." (**Lines 279-283**)

11. Line 279: GG medium: it is still unclear from the reference and this manuscript about the recipe. It appears acetate, formate and VFAs are added as standard (according to ref 41)? And no mention of rumen fluid.

Authors 'answer: The authors apologize for the mistake. SAB medium was our reference medium for methanogens culture, and we aimed to optimize it. Thereby, GG medium was derived from SAB medium with four differences as published in the patent FR 23 01404: acetate and formate concentrations were increased from 1g/L to 2g/L, H₂/CO₂ atmosphere was replaced by a simple nitrogen atmosphere in GG medium and VFAs were removed. We then decided to add ethanol as it was required for *Methanosphaera* sp. WGK6 isolation in a previous study.

We modified the paragraph as follow:

"A 200- μ L volume of this suspension was inoculated into a Hungate tube containing GG medium (29, patent FR 23 01404) that is derived from SAB medium (42), containing acetate, formate and methanol as standard, previously degassed for three minutes with 2-bar nitrogen." (**Lines 287-291**)

12. Line 287: if the GG medium was only supplemented with ethanol this does not align with the substrate analysis results. Did the isolation medium also contain rumen fluid and VFA? There is no reference of rumen fluid in either ref 41 or the SAB ref. In fact I wonder if acetate/format were present and ethanol was added as an electron donor at time of isolation was it using this as the carbon donor? The substrate analysis does not answer this question. Can BB6 use something other than methanol with ethanol? The results would suggest this is a strong possibility, it is an exciting observation, but it has not been addressed at all by the authors.

Authors 'answer: The authors apologize for the misunderstanding and hope that the answers to the previous comments have clarified this point (**Lines 171-177**). The final isolation medium was GG medium (containing acetate, formate and methanol) supplemented with ethanol. The substrate affinity analysis showed that BB6 can use methanol/ethanol or methanol/H₂/CO₂ in a basal GG medium (GG0 medium) that does not contains acetate, formate, fatty volatil acids and rumen. These results are consistent with those obtained for *Methanosphaera* sp. WGK6. Acetate and formate do not seem necessary but we keep them

as carbon source as the medium aims to be versatile, able to isolate several methanogen species.

13. Line 299: what is the COS agar recipe this has not be described?

Authors 'answer: The paragraph was modified as follow: "inoculation of 100 μ L on Columbia agar 5% sheep blood (COS) media (bioMérieux, Marcy-Étoile, France) plate." (Lines 309-310)

14. Line 414: authors mention rumen fluid, but it has not been described anywhere else in the methods. What breed was it collected from, what was it eating and how was this prepared for the medium, what % was added?

Authors 'answer: The authors apologize for the mistake; no rumen was added into the medium.

15. Line 419: remove "with methanol (2% v/v) or ethanol (2% v/v)" this looks like a repeat from lines 416-417?

Authors 'answer: The sentence was modified as demanded by the reviewer. (Line 424-430)

16. How did authors determine the CFU/mL? is this based on optical density? Please report.

Authors 'answer: To quantify the concentration in CFU/mL, we employed the established method utilizing the McFarland standard and measured the optical density using a spectrophotometer.

We modified the sentence as follow: "A calibration curve was obtained using serial dilution from 10^8 CFU/mL (0.5 McFarland turbidity standard) to 10^1 CFU/mL and was used to quantify the load of *Ca. M. massiliense* in each sample." (Lines 464-466)

17. Figure 1: gram stain image is not super clear; resolution could be improved.

Authors 'answer: According to reviewer' comment, the authors have improved the resolution of **Figure 1**.

18. Figure supp data 1: please include in this figure the addition of substrates for isolation (i.e. ethanol) without reading the legend the figure would imply you isolated BB6 on nitrogen alone. The picture of the agar plate still is not very clear, were authors able to pick a single colony? Was this spread or streaked on agar during the isolation process? A close up pic of a colony might be a better inclusion for this image so readers can see what it looks like morphologically?

Authors 'answer: We modified the figure **Figure S1** as demanded by the reviewer. BB6 appeared as a very thin translucent carpet, it was thus impossible to take a single colony and very difficult to take a good quality picture, particularly in the anaerobic chamber. Moreover, we didn't manage to transfer BB6 from solid GG medium to another GG medium plate, only subculture in GG broth medium was successful.

Reviewer 4 :

1) In this study, a *Methanosphaera* species was isolated using the GG medium, which was originally designed with formate and acetate as substrates and specifically intended for *Methanobrevibacter smithii*. According to the substrate affinity test, this *Methanosphaera* isolate grows on methanol under H₂/CO₂/N₂ atmosphere. However, it is unclear why methanol was not used in the initial isolation procedure. Can you provide clarification on this matter? You have only mentioned a reference for the GG medium. Briefly mention the major ingredients of this GG medium. It will be helpful for the readers.

Authors' answer: We modified the figure **Figure S1** as demanded by the other reviewers and also to provide more information about GG medium.

We also clarified the results and materials and methods sections as follow:

“*Ca. M. massiliense* was able to grow on GG0 medium (basal GG medium) with methanol under H₂/ CO₂/ N₂ atmosphere, on GG0 medium with methanol and ethanol under nitrogen atmosphere and on GG medium (containing acetate, formate and methanol) supplemented with ethanol as the reference condition (Table S4). Proxi-growth curve of methane production and optical density from day 0 to day 7 are also available in Table S4.” **(Lines 171-177)**

“A 200-μL volume of this suspension was inoculated into a Hungate tube containing GG medium (29, patent FR 23 01404) that is derived from SAB medium (42), containing acetate, formate and methanol as standard, previously degassed for three minutes with 2-bar nitrogen.” **(Lines 287-291)**

“A basal GG medium (GG0) derived from the original GG medium but without acetate, formate nor methanol or ethanol was used, and each substrate or substrate combinations were added separately in the medium GG0. Nine conditions were tested using GG0 medium, six under nitrogen atmosphere: no substrate, acetate (2 g/L), formate (2 g/L), methanol (2% v/v), ethanol (2% v/v), methanol and ethanol (2% v/v) and three under 5% H₂/ 20% CO₂/ 75% N₂ atmosphere without any substrate, with methanol or with ethanol.” **(Lines 424-430)**

2) Was the confocal light microscope the sole method used to ascertain the purity of the culture? How was the potential contamination with other methanogen species? While autofluorescence and cocci morphology were utilized to confirm the purity of the isolate, it should be noted that all *Methanosphaera* species share these characteristics. How did you ensure that the culture exclusively contained the specific isolate?

Authors' answer: The authors thank the reviewer for his comment. The comprehensive analysis of 16S rRNA sequencing revealed the identification of *Ca. M. massiliense* while no identification of *M. stadtmanae* was observed. Moreover, *de novo* genome assembling didn't retrieve *M. stadtmanae* genome or at least *M. stadtmanae* 16 rRNA gene.

3) Based on the phylogeny derived from the whole genome sequences, the current isolate exhibited close clustering with rumen *Methanosphaera* species (RUG761, SHI1033). However, the prevalence of these species in rumen samples was not examined in this study. Additionally, it would be intriguing to investigate whether the inclusion of rumen fluid in the GG medium enhanced the growth of this isolate. This investigation would provide valuable insights into the adaptability of the isolate to its specific environment.

Authors 'answer: According ANI analysis, *Ca. M. massiliense* should be the cultured representative of *Methanosphaera* sp. RUG761 and Mb5. Also, as our RT-PCR system was able to detect these species *in silico*, it was impossible for us to determine their prevalence. The reviewer is right and the introduction of the rumen into our culture medium will be tested in a future study that will be conducted in our team.

4. The media used for the pH susceptibility, antibiotic susceptibility, and substrate affinity test lacked clarity. Please provide the information on the media and substrates used. It was mentioned in the text (lines 413-415), that basal media without acetate, formate, volatile fatty acids, and rumen fluid were used for the substrate affinity test. However, supplementary data 6 mentioned that GG medium was supplemented with Ethanol (containing volatile fatty acids and rumen fluid). Therefore, please provide clarification on the specific media and substrates used for each test.

Authors 'answer: We also clarified the results and materials and methods sections as follow:

“*Ca. M. massiliense* was able to grow on GG0 medium (basal GG medium) with methanol under H₂/ CO₂/ N₂ atmosphere, on GG0 medium with methanol and ethanol under nitrogen atmosphere and on GG medium (containing acetate, formate and methanol) supplemented with ethanol as the reference condition (Table S4). Proxi-growth curve of methane production and optical density from day 0 to day 7 are also available in Table S4.” (Lines 171-177)

We also modified **Table S4**: “GG medium supplemented with Ethanol (contains acetate, formate, methanol)”

“A 200-μL volume of this suspension was inoculated into a Hungate tube containing GG medium (29, patent FR 23 01404) that is derived from SAB medium (42), containing acetate, formate and methanol as standard, previously degassed for three minutes with 2-bar nitrogen.” (Lines 287-291)

“A basal GG medium (GG0) derived from the original GG medium but without acetate, formate nor methanol or ethanol was used, and each substrate or substrate combinations were added separately in the medium GG0. Nine conditions were tested using GG0 medium, six under nitrogen atmosphere: no substrate, acetate (2 g/L), formate (2 g/L), methanol (2% v/v), ethanol (2% v/v), methanol and ethanol (2% v/v) and three under 5% H₂/ 20% CO₂/ 75% N₂ atmosphere without any substrate, with methanol or with ethanol.” (Lines 424-430)

5. What was the rationale behind selecting a 7-day time frame for assessing methane formation? Additionally, it would be beneficial to include growth curves for each 2 substrate. This would provide a more comprehensive understanding of the temporal dynamics and trends associated with growth/methane production.

Authors 'answer: We selected a 7-day time frame as the methane peaks at day 7. The authors provided a growth follow-up curve of methane production and optical density in **Table S4** and modified the text accordingly: “Proxi-growth curve of methane production and optical density from day 0 to day 7 are also available in Table S4.” (Lines 175-177)

6. pH susceptibility test: The authors conducted the test using a wide range of pH values (2-10), but specific pH requirements were not clearly defined in the current study.

Authors 'answer: In our study, we indeed conducted a pH susceptibility test by exposing the methanogen to a wide range of pH values ranging from 2 to 10. While we did not explicitly define specific pH requirements in the current study, our primary objective was to identify an optimal medium for the growth of the methanogen. Based on our experimental findings, we determined GG medium with a pH at 7.3 is suitable for the methanogen growth.

7. Why didn't you test the optimal temperature range for the growth of this isolate?

Authors 'answer: *Ca. M. massiliense* was isolated from a human stool sample and considering that the typical human body temperature is around 37°C, similar to the optimal growth temperature for other methanogens found in humans, we did not deem it necessary to test a temperature range. In comparison, the original article on *M. stadtmanae* also mentions a culture temperature range of 36°C to 40°C, while kangaroos exhibit a body temperature range of 35.5 to 36.9°C, and pigs have a range of 36°C to 39°C. Additionally, our laboratory lacks the necessary equipment for conducting such temperature manipulations.

8. Another significant limitation of this study is that much of the discussion surrounding the results appear to be more speculative rather than based on concrete evidence. The lack of appropriate references to support these speculations is also a notable drawback.

See below:

Lines 222-225- Methanogens primarily inhabit the colon of human GIT. Methanol derived from pectin fermentation serves as the primary substrate for Methanosphaera. The claim made about presence of methanol from beverages and food reaching the colon requires clarification. Are you referring to the levels of methanol/ethanol in the bloodstream here?

Authors 'answer: Methanol is possibly present in beverages, we added references to clarify this purpose and modified the text as follow:

"The enzymes encoded by genome sequencing may help define the population susceptible to *Ca. M. massiliense* implanted in the gut microbiota, which are defined as individuals exposed to ethanol and methanol produced by bacteria and fungi or from beverages and foodstuffs (31–33). Indeed, ethanol and methanol are both chemical compounds found in various sources that can be consumed by humans. Ethanol is primarily found in alcoholic beverages such as beer, wine, and spirits (34). Additionally, ethanol can be used as an additive or solvent in certain foodstuffs (35). Methanol can be present in foods due to natural fermentation processes or the breakdown of pectin in certain fruits and vegetables (33). Alcoholic beverages containing ethanol may also contain low levels of methanol, although these are generally considered safe for human consumption (31,32)." **(Lines 233-240)**

Lines 230-242: It has been proposed in this study that the methanogens isolated could potentially transfer from other mammals based on their prevalence in the feces of mammals. However, relying solely on prevalence is insufficient to support this suggestion. Methanogens are strictly anaerobic organisms, and their ability to contaminate food materials requires clear logical explanation. Gut microbes are known to be highly specific, making it difficult for other microbes to colonize easily. Even if they manage to colonize,

their ability to persist in the gut for an extended period of time is limited. The fact that the same isolate was obtained from the same individual at two different time points suggests that it has successfully colonized the gut. Further clarification is needed to support the claim of a potential transfer from other mammals.

Authors 'answer: While some archaea species are shared between animals and humans, our study is notable for the higher prevalence of *Ca. M. massiliense* in pigs and kangaroos compared to humans. In our study, only one human tested positive for the new methanogen, which is consistent with the findings of Chibani et al., where only one positive sample was identified out of over a thousand. The lower prevalence of *Ca. M. massiliense* in humans, compared to animals, suggests a potential for accidental transfer, possibly through food. Our strong belief is that *Ca. M. massiliense* was transferred to humans within a nourishing and protective bacterial biofilm, particularly against oxygen exposure.

The paragraph was simplified as demanded by the other reviewer to avoid speculation, we hope it will suit your requirements as well: "Indeed, through the results obtained in this study, we hypothesized that sources for *Ca. M. massiliense* may be mammalian zoonotic as suggested by the measured prevalence of *Ca. M. massiliense*. Food and other animal contacts might be a source for host transmission of microorganisms to the intestinal microbiota of humans (32)." **(Lines 245-240)**

Specific comments

- Line: 96: what is the recently designed hydrogen-free carbon-dioxide-free medium: Give a reference or the name of the media here.

Authors 'answer: We added the name of the medium and the corresponding reference and the patent number as follow: « Using a recently designed hydrogen-free and carbon dioxide-free medium named GG medium (29, patent FR 23 01404) » **(Lines 97-99)**.

- Line 315: culture or culture supernatant?

Authors 'answer: The word supernatant was removed **(Line 317)**.

- Line 414: volatile fatty acids and rumen fluid

Authors 'answer: GG medium as published in the patent FR 23 01404 doesn't contain rumen and fatty volatile acids.

The paragraph was modified as follow: "A basal GG medium (GG0) derived from the original GG medium but without acetate, formate nor methanol or ethanol was used, and each substrate or substrate combinations were added separately in the medium GG0. Nine conditions were tested using GG0 medium, six under nitrogen atmosphere: no substrate, acetate (2 g/L), formate (2 g/L), methanol (2% v/v), ethanol (2% v/v), methanol and ethanol (2% v/v) and three under 5% H₂/ 20% CO₂/ 75% N₂ atmosphere without any substrate, with methanol or with ethanol." **(Lines 424-430)**

- Figure 1: this figure especially the electron microscopy figures B and C are not clear enough to observe specific features of the cell.

Authors 'answer: The authors have improved the resolution of the **Figure 1**.

- Table 1: Can you expand the 2nd column length?

Authors 'answer: The **Table 1** was modified as demanded.

- Table S2: Mention the 16S rRNA gene sequence in the title.

Authors 'answer: The Table S2 title was modified as follow in the text (**Line 720-722**) and in the Table S3 excel file: "Table S2: *Candidatus* Methanosphaera massiliense sp. nov. 16S rRNA gene sequence and the first 100 hit blasts downloaded from NCBI GenBank Database (03-01-2023)."

Reviewer 5:

The main point of the manuscript from Pilliol et al is the isolation (but relatively poor description) of a new species from the genus *Methanosphaera*, obtained from a human stool sample. The importance and diversity of archaea in humans, and in human gut microbiota (mainly methanogens) has been highlighted by several and sometimes important reviews on this topic. Having cultured representatives besides (meta)genomic data is an important step for addressing physiological or ecological questions.

However, I feel frustrated after reading and analysing this manuscript (the revised version is the only version that I have reviewed), concerning several data given by authors. This is described in more details below.

Authors 'answer: The authors would like to thank the reviewer for his valuable comments. The authors hope that the new manuscript will give entire satisfaction to the reviewer and that he will take in consideration that methanogens culture and characterization are very fastidious tasks.

Of all, some elements are not clear by lack of details and the title sounds mysterious to me as it is said that this isolate "questions transfer between human and animal microbiota": This seems of course new about this isolate, but not really new compared to some other and sometimes old papers (eg. archaeal DNA / sequences, even organisms seen from human studies like Haloarchaea, other orders of methanogenic archaea and other species from Methanobacteriales which are found in ruminants, pigs, ...). This could be reasonably forgotten if the manuscripts provided more specific data to give some other clues supporting a transfer from animal / foodstuff rather than a unique human subject and prevalence in some animals.

Authors 'answer: While some archaea species are shared between animals and humans, our study stands out due to the higher prevalence of *Ca. M. massiliense* in pigs and kangaroos compared to humans. In our study, only one human tested positive for the new methanogen, and a similar finding was reported in Chibani et al.'s study, where only one out of over a thousand samples was positive. The lower prevalence of *Ca. M. massiliense* in humans, compared to animals, suggests a potential for accidental transfer, possibly through food. However, it is important to note that this hypothesis cannot be generalized based solely on the similarity in prevalence between animals and humans for other methanogen species. While a closer prevalence match may indicate alternative transmission routes, it is crucial to consider other factors such as host-specific adaptations, ecological niches, and potential reservoirs, which can vary significantly among different methanogen species. Therefore, further investigations and comprehensive analyses are needed to understand the transmission dynamics and potential cross-species transfers of methanogens between animals and humans.

Some of these points are detailed below. It is of course very likely that this isolate belongs to a host-associated group and not an environmental one (like other *Methanosphaera* spp and as it can be inferred from Supp Data 3 in which no «real» environmental sequences are retrieved from the first 100 hit blasts - this is an important point for readers that could be clearly disclosed in the manuscript).

Authors 'answer: We modified the sentence to provide more information's in the manuscript:

“The 16S rRNA sequence-based phylogenetic tree showed that the new methanogen clustered with only host-associated sequences and not with environmental ones, it also firstly clustered with uncultured archeon clones from pig faeces (accession n°: HM573447.1 and HM573413.1) and *Methanosphaera* sp. WGK6 16S rRNA (accession n°: KF697728.1) (Table S2 and figure 3).” (Lines 131-136)

Also, I strongly recommend authors to send their isolate to a second collection of organisms (ATCC, DSMZ, JCN, ...) to ensure it to be safe among time.

Authors 'answer: Our isolate is in preparation to be sent to a second collection.

Specific comments and questions arising from the reviewing:

1. The isolation relies on a new medium called "GG medium" : it has been recently published by the author (ref 41 in the manuscript) and indicated as patented ("conflict of interest section"). As its publication is recent, it would be interesting to give its composition in the manuscript, either in M&M section or added to a figure legend (eg legend of Supp Data 1). Moreover, as I could not retrieve the recipe from ref 41, it is very unclear of what is present and at which quantity within this growth medium: eg, does it contain volatile fatty acids and filtered rumen fluid as suggested by supp data 6 "substrate affinity test" ? If yes, which ones ? It could be important for the reader to know, considering the results observed for substrate affinity tests.

Authors 'answer: We modified the figure **Figure S1** as demanded by the reviewer and also by the second reviewer to provide more information about GG medium.

As also demanded by the second reviewer, we clarified the results and materials and methods sections as follow:

“*Ca. M. massiliense* was able to grow on GG0 medium (basal GG medium) with methanol under H₂ / CO₂ / N₂ atmosphere, on GG0 medium with methanol and ethanol under nitrogen atmosphere and on GG medium (containing acetate, formate and methanol) supplemented with ethanol as the reference condition (Table S4). Proxi-growth curve of methane production and optical density from day 0 to day 7 are also available in Table S4.” (Lines 171-177)

“A 200-μL volume of this suspension was inoculated into a Hungate tube containing GG medium (29, patent FR 23 01404) that is derived from SAB medium (42), containing acetate, formate and methanol as standard, previously degassed for three minutes with 2-bar nitrogen.” (Lines 287-291)

The name « GG0 medium » was introduced in the materials section and referred to the basal GG medium used in this study:

“A basal GG medium (GG0) derived from the original GG medium but without acetate, formate nor methanol or ethanol was used, and each substrate or substrate combinations were added separately in the medium GG0. Nine conditions were tested using GG0 medium, six under nitrogen atmosphere: no substrate, acetate (2 g/L), formate (2 g/L), methanol (2% v/v), ethanol (2% v/v), methanol and ethanol (2% v/v) and three under 5% H₂ / 20% CO₂ / 75% N₂ atmosphere without any substrate, with methanol or with ethanol.” (Lines 424-430)

o About the isolation and description of this new species, it would be great to conform to

some standards. The growth optimal pH is defined at 7.3, only because there is a growth at pH6 and pH7.3 and not at pH5.0 and pH8.0, which does not seem sufficient to claim 7.3 as optimal pH.

Authors 'answer: The authors agree with the reviewer, pH=7.3 is the reference pH for GG medium. We modified the sentence accordingly: "The pH of 7.3 in the GG medium was considered as the reference pH for the methanogen culture." (Lines 168-169)

o What is the sensitivity to O₂, and micro-aerophily of this isolate, its viability in aerobia environment ? Could you test it in order to give any support to your hypothesis of transfer from foodstuff ?

Authors 'answer: The authors thank the reviewer for the comment and according to the comment, the authors have analyzed the *M. massiliense* genome looking in for enzyme that could support micro-aerophilic lifestyle and the authors attest that *Ca. M. massiliense* genome does not encode for enzyme that could support micro-aerophilic lifestyle. However, as methanogens have a symbiotic lifestyle, our hypothesis is that the transfer occurs in a protective biofilm with other aerophilic or aerotolerant bacteria. Furthermore, the authors exposed *Ca. M. massiliense* to ambient air for 5 minutes and observed that these 5 minutes were enough to kill *Ca. M. massiliense* according to methane measurement (No exposure, T₀ : AUC = 0, T₂ (day 7) : AUC = 6310698,667 / 5 minutes exposure, T₀ : AUC = 0, T₂ (day 7) : AUC = 0).

2. Parts about the human subject who is positive for the isolate are very unclear for me and has to be clarified:

o It is written lines 189-191 that "the one positive human faeces sample had been collected from the same individual from whom *Ca. M. massiliense* had been isolated by culture as above". This means that a 2nd collect of faeces has been done? Could you indicate, even approximately, the time passed between the two sampling dates (days, weeks or months ?) in order to sustain a gut-inhabiting microbe in this subject? Or is it a miswording as it is written in M&M "specimen collection" part, that anonymized leftovers were used.

Authors 'answer: The authors apologize for the misunderstanding. There was only one stool sample. We modified the sentence as follow : "The one positive human faeces sample was the same whom *Ca. M. massiliense* had been isolated by culture as above." (Lines 201-202)

We are not able to obtain a second stool sample from the same individual as sample is anonymized in this study.

o In fact, it has to be clearly established for the reader if this species has been retrieved twice independantly from the same donor or not. If yes, isn't it possible (ethically) to give at least her/his age/gender/ethnicity? Ideally also, other informations like health status and profession.

Authors 'answer: Unfortunately, as we used a single anonymized leftover stool sample, it is not possible to provide this information legally in a publication.

3. Detection of the isolate has been performed in stools of several animals. From data given in Supp Data 13 :

o while the qPCR detection test seems ok as it is described and how it was implemented, how can be explained such a discrepancy between Ct values and CFU/mL observed in the human subject compared to all other positive animals ? Eg, the CFU values in animals would theoretically lead to Ct values about 24-25 (and not 32-33 as observed), if compared to the values observed in the unique human. In fact, I am wondering about the real quantity / proportion of this isolate in the human gut microbiota. Has the qPCR assay a good linearity over the quantities which are tested ?

Authors 'answer: Initially, we were looking for positive/negative sample without quantification. The manipulation was redone by precisely weighing the stool sample (0.5 g in 500 µL of G2 buffer, **Lines 439-441**), we modified the sentence accordingly: "Moreover, *Ca. M. massiliense* load was comparable in the human sample (3,77E+07 CFU/mL) and in pig and kangaroo samples (average load 2,68E+07 CFU/mL +/-1,25E+07) (Table S6a)." (**Lines 206-208**)

o these data clearly indicate that this species is far more prevalent in red Kangaroos and pigs (and not present in other animals that were tested) but far less in quantity in all these animals than in the unique human. Could you indicate approximately to which proportion it corresponds among other faecal organisms, in this human subject, compared to other animals ? In the human subject, is it associated with other methanogens?(at which comparative abundance and which ones ?)

Authors 'answer: As the new manipulation showed that the methanogen load was comparable in all the positive samples, we did not further pursue the investigations, but the authors agree with the reviewer that studying the proportion relative to other bacterial species present would be necessary. However, this would require additional metagenomic analyses that we were unable to perform for now.

4. besides the isolation of this *Methanospheara* sp. strain, the authors highlight that this occurrence in a human subject arises likely from a transfer from animal (pigs, less likely kangaroos, as it was not detected in any other animal tested, ie dogs, cows, sheep and horses), and that could occur from foodstuff. Even if such a zoonotic transfer hypothesis is likely, I would omit the text from lines 230 to 242, just keeping "Indeed, through.... As suggested by prevalence of *Ca. M. massiliense* in some animals." (I would however keep the last paragraph of the conclusion as it stands). Lines 232 to 240 seem to me out of the discussion.

Authors 'answer: The paragraph was modified as demanded by the reviewer: "Indeed, through the results obtained in this study, we hypothesized that sources for *Ca. M. massiliense* may be mammalian zoonotic as suggested by the measured prevalence of *Ca. M. massiliense*. Food and other animal contacts might be a source for host transmission of microorganisms to the intestinal microbiota of humans (32)." (**Lines 245-249**)

We kept the second sentence to support our hypothesis.

Other minor points:

- abstract :

o please change line 35 "...and 0.67% in human faeces" by "and no detection in 149 (or 150 ?) other human samples"

Authors 'answer: The sentence was modified as demanded: "Screening additional mammal and human faeces using a specific genome sequence-derived DNA-polymerase RT-PCR system yielded a prevalence of 22% in pig, 12% in red kangaroo and no detection in 149 other human samples." (Lines 35-36)

o the word «origin» line 36 is ambiguous. Please rephrase.

Authors 'answer: We changed the word "origin" by the word "sources" (Line 37).

- depending on answers to point 2 and 3, please remove "and adaptation to human gut" (line 45) as evidences seem lacking to claim that this species is inhabiting the gut of this human subject.

Authors 'answer: We removed "and adaptation to human gut" as demanded. (Line 45)

- Line 429 and after: please indicate in the text if in silico, the qPCR test should be positive for strain RUG761 (and also strain Mb5)?

Authors 'answer: We added the testing protocol in the materials and methods section:

"RT-PCR specificity was in silico tested by the NCBI BLAST program (<http://www.ncbi.nlm.nih.gov/BLAST>) using each primer alone and in-silico amplified using both primers and the specific probe, particularly against *Methanosphaera* sp. Mb5, RUG761, SHI1033, WGK6 genome sequences, using Amplify4 software (version 1.0, Bill Engels, University of Wisconsin)." (Lines 450-454)

The results were added in the appropriate section:

"Moreover, the RT-PCR system was *in silico* positive for *Ca. M. massiliense*, *Methanosphaera* sp. Mb5 and RUG761 but not for *Methanosphaera* sp. SHI1033 and WGK6." (Lines 192-194)

- Line 223 "implanted" instead of "implanting"?

Authors 'answer: The sentence was modified as demanded (Line 231).

- Lines 230-235: please rephrase and omits numerical values as it does not make sense with only one positive human sample.

Authors 'answer: The paragraph was modified as demanded above by the reviewer, the numerical values do not appear anymore: "Indeed, through the results obtained in this study, we hypothesized that sources for *Ca. M. massiliense* may be mammalian zoonotic as suggested by the measured prevalence of *Ca. M. massiliense*. Food and other animal contacts might be a source for host transmission of microorganisms to the intestinal microbiota of humans (32)." (Lines 245-249)

- Line 685, "calculated FROM THE using PYANI...."

Authors 'answer: The sentence was modified as follow: "Heatmap phylogeny of *Candidatus Methanosphaera massiliense* sp. nov. and closely related methanogen species generated

with ANI values calculated using PYANI software version (0.2.7) with standard parameters”.
(Lines 686-688)

Re: Spectrum05141-22R2 (*Candidatus* Methanosphaera massiliense sp. nov. questions transfer between human and animal microbiota.)

Dear Prof. Elodie Terror:

Thank you for the privilege of reviewing your work. Below you will find my comments, instructions from the Spectrum editorial office, and the reviewer comments.

We thank the authors for their hard work to improve the manuscript. There are still primarily text edits required (please see reviewer comments), but then the manuscript can be accepted.

Revision Guidelines

Sincerely,
Henning Seedorf
Editor
Microbiology Spectrum

Reviewer #4 (Comments for the Author):

Dear authors,
The revised manuscript has addressed the majority of the comments. However, there are a few minor points that require attention:

- In the discussion section, it is recommended to provide compelling arguments justifying the statement about the "transfer between human and animal microbiota."
 - Please incorporate details about the method used to measure optical density.
 - Regarding lines 227-230, the current sentence lacks clarity. Consider revising it to make it more clear.
- Thank you.

Reviewer #5 (Comments for the Author):

First, thanks for your interesting work and to have taken into account most of my initial remarks. Some specific remarks are indicated below.

Your rebuttal about one major point does not answer clearly to one of my main concern. Horizontal transmission of gut microbes, zoonotic, foodstuff-originating,... and the implantation of new inhabitants during the lifespan is a general and important / central question so that your title seems not adequate ('...questions transfer between human and animal microbiota ») as your results do not imply some new elements that would strengthen one hypothesis or even making others more unlikely. Currently, your result indicates that your isolate belongs to a species that is recovered from several animals (eg pigs and red kangaroos as shown by your work, rumen cow, sheep, and also human). There is no evidence of lineages that would be specific of one non-human host from which would originate your isolate. Therefore I would strongly suggest that your title was turned into a more conventional one like « *Cand M. massiliense* sp. nov., a methanogenic archaeal species isolated from a human fecal sample and being prevalent in pigs and red kangaroos », reflecting more precisely the findings.

Other minor points and typos:

- Line 137, typo : « Methanospheara » Lines 140-143 : please indicates the animal origin of the archaeal genomes M5, RUG761 and SHI1033 in the text (eg « ...SHI1033 from sheep »)
- Line 218, typo : « which » seems to have to be deleted
- does genome SHI1033 (line 143) correspond to the one mentioned line 90 from Chibani et al and to the one ? If yes, please make a link so that the reader will know.
- Please mention the name of the other public collection to which your isolate has been sent, even if there is no accession number currently.

Dear authors,

The revised manuscript has addressed the majority of the comments. However, there are a few minor points that require attention:

- In the discussion section, it is recommended to provide compelling arguments justifying the statement about the "transfer between human and animal microbiota."
- Please incorporate details about the method used to measure optical density.
- Regarding lines 227-230, the current sentence lacks clarity. Consider revising it to make it more clear.

Thank you.

The authors would like to thank the reviewers for their valuable and pertinent comments. They hope this revised version will give to the reviewers entire satisfaction and could be acceptable for publication.

Reviewer #4 (Comments for the Author):

Dear authors,

The revised manuscript has addressed the majority of the comments. However, there are a few minor points that require attention:

- In the discussion section, it is recommended to provide compelling arguments justifying the statement about the "transfer between human and animal microbiota."

Author's answer: In response to the reviewer's comment, the authors have provided clarification regarding the absence of evidence supporting the transfer of microbiota between animals and humans, as discussed in the newly added paragraph within **lines 246 to 263**. Consequently, they have heeded the reviewer's suggestion and modified the title of the study accordingly.

- Please incorporate details about the method used to measure optical density.

Author's answer: The authors provided details about the method used to measure optical density. The text was modified as follow: "optical density was measured by putting the Hungate tubes in a spectrophotometer cell density meter model 40 (ThermoFischer Scientific, Waltham, Massachusetts, United States)" (**Lines 299-301**).

- Regarding lines 227-230, the current sentence lacks clarity. Consider revising it to make it more clear.

Author's answer: The sentence has been rephrased accordingly **lines 230-235**.

Reviewer #5 (Comments for the Author):

First, thanks for your interesting work and to have taken into account most of my initial remarks. Some specific remarks are indicated below.

Your rebuttal about one major point does not answer clearly to one of my main concern. Horizontal transmission of gut microbes, zoonotic, foodstuff-originating,... and the implantation of new inhabitants during the lifespan is a general and important / central question so that your title seems not adequate ('...questions transfer between human and animal microbiota ») as your results do not imply some new elements that would strengthen one hypothesis or even making others more unlikely. Currently, your result indicates that your isolate belongs to a species that is recovered from several animals (eg pigs and red kangaroos as shown by your work, rumen cow, sheep, and also human). There is no evidence of lineages that would be specific of one non-human host from which would originate your isolate. Therefore, I would strongly suggest that your title was turned into a more conventional one like « **Cand M. massiliense sp. nov., a methanogenic archaeal species isolated from a human fecal sample and being prevalent in pigs and red kangaroos** », reflecting more precisely the findings.

Author's answer:

We thank the reviewer for his feedback, which prompted the author to modify the title as suggested (**Lines 1-3**).

Other minor points and typos:

- **Line 137, typo:** « **Methanospheara** »

Author's answer: The word was corrected accordingly (**Line 138**).

- **Lines 140-143:** please indicates the animal origin of the archaeal genomes M5, RUG761 and SHI1033 in the text (eg « ...SHI1033 from sheep »

Author's answer: The sentence was modified as follows:

“However, it clustered with *Methanosphaera* sp. RUG761 **from bovine** (ANI, 99%) and *Methanosphaera* sp. M5 **from human with colorectal cancer** (ANI, 99%), two genomes that clustered together (ANI, 99%), and with *Methanosphaera* sp. SHI1033 **from sheep** (ANI, 97%) (Figure 4).” (**Lines 143-146**)

- **Line 218, typo:** « **which** » seems to have to be deleted

Author's answer: The word “which” was deleted accordingly (**Line 221**).

- **does genome SHI1033 (line 143) correspond to the one mentioned line 90 from Chibani et al and to the one? If yes, please make a link so that the reader will know.**

Author's answer: The genome SH1033, retrieved from sheep, does not correspond to one retrieved by Chibani *et al.* but was mentioned in their phylogenetic analysis.

- Please mention the name of the other public collection to which your isolate has been sent, even if there is no accession number currently.

Author's answer: The authors clarify that the isolate was already deposited in international public collection CSUR WDCM 875 (Collection Souches Unité des Rickettsies, WDCM 875, Marseille, France) under the reference Q7470 and sent to the DSMZ.

This point is clarified in **lines 152-153**.

Re: Spectrum05141-22R3 (*Candidatus Methanosphaera massiliense* sp. nov., a methanogenic archaeal species found in a human fecal sample and being prevalent in pigs and red kangaroos.)

Dear Prof. Elodie Terror:

Your manuscript has been accepted, and I am forwarding it to the ASM production staff for publication. Your paper will first be checked to make sure all elements meet the technical requirements. ASM staff will contact you if anything needs to be revised before copyediting and production can begin. Otherwise, you will be notified when your proofs are ready to be viewed.

Sincerely,
Henning Seedorf
Editor
Microbiology Spectrum